# ConDABench: Interactive Evaluation of Language Models for Data Analysis

## Abstract

Real-world data analysis tasks often come with under-specified goals and unclean data. User interaction is necessary to understand and disambiguate a user's intent, and hence, essential to solving these complex tasks. Existing benchmarks for evaluating LLMs on data analysis tasks do not capture these complexities or provide first-class support for interactivity. We introduce **ConDABench**, a framework for (i) generating conversational data analysis (**ConDA**) benchmarks and (ii) evaluating external tools on the generated benchmarks. **ConDABench** consists of (a) a multi-agent workflow for generating realistic benchmarks from articles describing insights gained from public datasets, (b) 1,420 ConDA problems generated using this workflow, and (c) an evaluation harness that, for the first time, makes it possible to systematically evaluate conversational data analysis tools on the generated ConDA problems. Interestingly, evaluation of state-of-the-art LLMs on these ConDA benchmarks reveals that while the new generation of models are better at solving more instances, they are not necessarily better at solving tasks that require sustained, long-form engagement. Hence, **ConDABench** can be an avenue for model builders to measure progress towards truly collaborative models that can complete complex interactive tasks.

## 1 Introduction

LLM-powered conversational assistants such as ChatGPT and Gemini are rapidly gaining popularity for supporting a wide range of cognitive tasks. One area seeing especially notable growth is data analysis, with applications expanding across domains like business intelligence  (Vidgof et al., 2023), healthcare  (Harrer, 2023), and scientific research  (Boiko et al., 2023; Low & Kalender, 2023). Despite their growing use, LLMs remain imperfect at performing data analysis due to the task's inherent complexity, which demands logical reasoning, code generation, procedural execution, and interactivity. These challenges make data analysis a particularly valuable test bed for evaluating the capabilities and limitations of LLMs. However, generating realistic benchmarks capturing the nuances of real-world data analysis is challenging. User queries are often *vague and underspecified*, particularly with respect to the underlying data context. Users frequently refine queries iteratively while *interacting* with a data analysis assistant. Furthermore, real-world datasets are often unclean, which needs to be reflected in the benchmark set.

We introduce **ConDABench**, a suite of tools to **generate** benchmark sets and to effectively **evaluate** Conversational Data Analysis (ConDA) capabilities of modern assistants. The first component of **ConDABench** is a **modular, multi-agent benchmark generation framework** for generating diverse and challenging data analysis problems (Fig. 1). This modular approach is essential for capturing the heterogeneity of real-world analytical workflows, allowing us to curate evaluation problems spanning: (a) **Open-ended** queries, where models must explore multiple possible interpretations of an analysis problem; (b) **Projection** queries, where models must perform forecasting; (c) **Traditional question-answering** problems, where models must find a well-defined answer based on provided data (examples in Appendix B). We use the above framework to generate a specific dataset, also called **ConDABench**, especially designed to assess LLMs in the context of conversational data analysis on real-world analysis problems. Finally, the third piece of the framework is a **benchmark evaluation harness** that can be used to evaluate interactive data analysis tools on **ConDABench**.

**Key challenges:** *How to create benchmarks that can support the automated evaluation of human-in-the-loop interactive tools?* In existing benchmarks for evaluating interactivity, such as in MT-

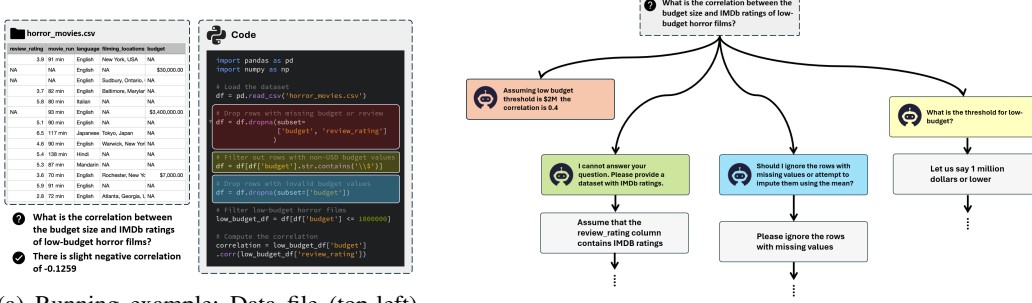

Figure 1: **ConDABench Generation.** From an article and data $d$, Curation pipeline extracts a query-answer pair $(q, a)$ for which the code generation pipeline produces code $c$ to support answer $a$. Details on agents' implementation and prompt outlines can be found on `https://github.com/condabench/condabench_details`.

(a) Running example: Data file (top-left), query-answer pair (bottom-left) and associated supporting code (right).

(b) A tree of possible conversations when a data analysis assistant is presented with the query from Figure 2a.

Figure 2: Running Example and a tree of conversation trajectories

BENCH (Zheng et al., 2023b; Bai et al., 2024), followup questions are initiated by the user and are static and part of the benchmark. This might not be realistic since data analysts condition their followups based on LLM's output. In addition, data analysts also *answer* questions that a conversational LLM agent may pose. How can we automate this conversation and evaluate it without human-in-the-loop? A second challenge when generating an interactive benchmark is ensuring the correctness and quality of the benchmark.

We address both of these challenges by including *code* along with the data files, the query, and the answer. A key component of our multiagent framework for benchmark generation is a code generator that iteratively finds the code that correctly explains the answer given the query and data files. The code represents the *latent reasoning* required to generate the answer from the query. The code serves two purposes. First, it provides grounding for the query-answer pair, thus ensuring the quality of the benchmark (Fig. 1). Second, it is used to implement the *User Proxy Agent*, which leverages the code to automatically answer queries that a data analysis tool may pose when given a vague or underspecified query. The User Proxy agent is a critical part of our **ConDABench** evaluation harness. The User Proxy answers questions (about the task) by referring to the code, but without revealing the code. The code allows the User Proxy to generate conversationally realistic answers for questions a data analysis tool might ask when analyzing the dataset.

We summarize the contributions of this paper as follows: (i) A **modular multi-agent architecture** for curation of *realisitc* data analysis benchmarks. (ii) A **code-grounded benchmark set** generated using

Table 1: Comparison to Other Benchmarks (More related work in Appendix A)

| Benchmark | # Datasets | # Queries | Conversational Eval | Open-Ended Queries | Free-form Answer Support | Multi Dataset Queries | Unclean Dataset |
|---|---|---|---|---|---|---|---|
| WikiTableQuestions | 2108 | 22033 | ✗ | ✗ | ✗ | ✗ | ✗ |
| InfiAgentDABench | 52 | 257 | ✗ | ✗ | ✓ | ✗ | ✓ |
| TAPilot-Crossing | 5 | 1024 | ✗ | ✓ | ✓ | ✗ | ✗ |
| **ConDABench** (ours) | 1855 | 1420 | ✓ | ✓ | ✓ | ✓ | ✓ |

the above framework, with **1420** queries grounded in **338** real-world data-analysis articles, which reflects the complexities of an analyst working over the data containing a diverse styles of queries. (iii) An **interactive evaluation harness** catered to automate evaluation of *conversational systems* using a carefully designed *User Proxy Agent*, and **systematic evaluation metrics** that measure both correctness and conversational quality of interactions. (iv) Detailed **analysis** on the conversationality and performance of various state-of-the-art LLMs (sample set on `https://condabench.github.io`).

## 2   Why is Evaluating Interactive Assistants Hard?

**Supporting interactivity in the evaluation harness.**  Consider the query at the top of Fig. 2b. A data analysis assistant may respond to the query in one of several ways as depicted. For example, it may respond with: (a) a direct answer to the query along with a statement of assumptions, (b) a question about data cleaning steps, or (c) a question about an analysis parameter. Now, the evaluation harness must continue the conversation, but different choices can alter the assistant's final answer. For the question about "low-budget" threshold, if the threshold used to compute the ground truth answer does not match the threshold provided by the evaluation harness, the final response produced by the assistant will be incorrect despite the assistant doing "everything right". Generating follow-up responses compatible with the ground-truth answer is the primary challenge facing interactive benchmarking of data analysis assistants.

Our first key idea is to **use code that produces the expected answer** as the grounding to generate these compatible responses. We dub this component of the evaluation harness that produces responses the **user proxy**. The code that produces the expected answer is shown in Fig. 2a, and based on that code, the evaluation harness can answer question saying that the threshold is $\$1,000,000$. The example in Fig. 2a is a modified version of a benchmark task automatically generated using the **ConDABench** pipeline.

**Sourcing realistic data-analysis tasks.**  The second major challenge in benchmarking interactive data analysis assistants is in sourcing realistic tasks where the answer is still verifiable. With recent advances in AI models, even complex multi-step tasks can be handled automatically (Martin Iglesias, 2025). Interactivity is relevant in specific situations, such as when the data is unclean, the task is ambiguous, or exploratory analysis is needed to fully define the scope of the analysis task. Previous attempts have been made to synthetically generate these types of tasks, for example, by artificially injecting ambiguity (Kim et al., 2024; Zhang et al., 2024), however, these still do not cover the complexity and the gamut of tasks requiring interaction.

Our second key idea is to **use real data analysis articles** (e.g., data journalism, documented notebooks, scientific literature, etc.) together with their source data to generate tasks, with the expected answers coming from the text inside the article (see Section 3.1). However, we are now left with another issue: these articles rarely provide the code that was used to produce the answer, nor do they describe the choices made for data cleaning, statistical tests, or analysis parameters. Given that these are needed to build the user proxy as described above, the second key challenge is to *reverse engineer* the analysis code from the data analysis article, query, and the ground-truth answer. In Fig. 2a, the code was automatically generated from the journalism article and the expected answer.

## 3   Benchmark Construction

A data analysis *problem* $t$ in our setting is given by $t = (q, a, d, c)$, where $q$ is a natural language *query*, $a$ is the *answer* to the query, $d$ is a collection of *data* files, and $c$ is the supporting *code* used to generate the answer. The answer $a$ can take various forms–an image plot, a dataframe, a single numerical value, or a natural language response. The code $c$ is a (Python) program that performs data analysis, starting from the data files $d$, to produce artifacts such as numerical answers or plots to substantiate the answer $a$. As in Fig. 1, the benchmark construction workflow consists of two steps: (a) query-answer extraction, and (b) code generation. In the following, we elaborate on each step.

### 3.1   Query-Answer Curation

The first step in the construction of our benchmark dataset is the extraction of query-answer $(q, a)$ pairs from various online sources. We start with sources that include both *articles* and public *datasets*, e.g., web pages, manuscripts, blogs, or Python notebooks that contain in-depth analyses of public

datasets. Starting from these published articles and datasets ensures that we focus on both complex, real-world analysis tasks on un-processed datasets grounded in human-conducted analysis.

We introduce a *Curator*, an LLM-based agent, to process these articles (potentially consisting of text, code snippets, and visualizations), and to produce query-answer pairs $(q, a)$. The Curator's task also includes generating different categories of queries for direct question-answering, open-ended, and projection tasks. We further use a *Reviewer* agent with a validation prompt to check if each query-answer pair is correctly supported by the original article.

***Running Example***. In a running example (Fig. 2a), the source article might have text similar to *"As opposed to other genres, the rating of low-budget horror movies have almost no relation to the budget itself with a very negligible correlation coefficient of $-0.1259$"*. From this, the Curator can generate the query-answer pair in the figure. Note that this query is inherently under-specified. Namely, the definition of "low-budget" is neither in the query nor in the article extract.

## 3.2 REVERSE ENGINEERING DATA ANALYSIS VIA CODE GENERATION

The second step starts with the $(q, a, d)$ tuple to generate the supporting code $c$. Note that this task is not the same as solving the data analysis problem since the answer $a$ is already known while generating the code $c$. The purpose instead is to *reverse engineer* the data cleaning, analysis techniques, and parameters that the author of the data analysis article used to arrive at the answer. The generated code does not need to output the answer $a$ verbatim, but only need to output sufficient numerical and visual artifacts to *support* the answer.

***Running Example***. In the running example from Fig. 2a, the code generation step crucially involves finding the threshold parameter for what movies are "low-budget". Picking the different values for the threshold will produce different correlation values.

The code generation pipeline consists of two interactive agents, the *Code Generator* and the *Reviewer*. The Code Generator is provided with only the query $q$ and the dataset $d$ (not the answer $a$) and is tasked to produce code $c$. The Reviewer is additionally provided with the answer $a$ and checks if the output of the code $c$ matches $a$ and if not, it generates feedback that is sent back to the Code Generator. This process continues iteratively until the code generator produces code which upon execution produces an answer that matches $a$. If this iterative process reaches a maximum bound without the appropriate code being generated, we deem that curated $(q, a)$ pair as incorrect and filter them out. Note that both the Code Generator and the Reviewer are allowed to execute code. The Code Generator-Reviewer interaction eventually produces code $c$ and hence, the full data analysis problem $(q, a, d, c)$. Code generation not only is useful to produce the code $c$ which can be used to support interactions discussed in Section 2, but also acts as a "proofing" step to ensure that the original data analysis article is correct.

***Running Example***. Fig. 3 shows a typical interaction between the Code Generator and Reviewer. Here, after addressing the missing values, the Code Generator could produce code with the wrong threshold for defining "low-budget". This code is executed and the Reviewer notices that the answer does not match the expected answer of $-0.1259$ and provides feedback that the correlation is higher than expected and that the Code Generator should try to vary the threshold. The Code Generator now regenerates the code with a different, correct threshold that produces the expected correlation value.

**Answer Leakage and Audited Reviewer.** Building a reviewer for the code generation step is not fully straight-forward. A naive reviewer is vulnerable to the problem of (partial) *answer leakage*. Consider a query "What is the average age in the department with the most faculty?" and an answer "The most populous department, history, has average age 49". If the code generator mistakenly determines psychology as the most populous department (potentially due to missing data cleaning steps), a naive reviewer might respond "The most populous department is history. Please revise your code". In the next iteration, the code generator may potentially hard-code "history" and compute the average age, i.e., `print(df[df.dept == "History"].age.mean())`, bypassing the actual task. We call this issue the answer leakage problem. The ideal feedback in the code generation step points the Generator in the right direction, but should not reveal the intermediate or final answer. To avoid this problem, we use a structured process using a series of tasks where (Appendix C): (a) the Reviewer agent first compares the output of the generated code against the answer $a$; (b) if they do not match, it produces feedback on the reason for the mismatch; (c) the it audits the feedback to check for answer leakage and the usefulness of the feedback before passing it back to the Code Generator.

Table 2: **Left:** Summary statistics of **ConDABench**. **Right:** Stepwise statistics for benchmark construction. *Curated* is the number of initial query-answer pairs, *Code Gen. Passed* have correct code and passed all checks.

| size | 1420 |
|---|---|
| #data files | 1855 |
| avg. #data files per query | 1.31 |
| % of queries taking > 1 data files | 17.25 |
| avg. data file size per query | 4.24MB |
| # viz. based queries | 405 |
| avg. code length | 24.02 |
| avg. code exe. time | 2.01 sec |
| avg. # libraries per code | 1.56 |
| **shallow** (tasks converging in <3 iters) | 1317 |
| **deep** (tasks converging in >= 3 iters) | 103 |

| Source | Articles | Type | Curated | Code Gen. Passed |
|---|---|---|---|---|
| TidyTuesday | 283 | qa | 1004 | 551 |
| | | open-ended | 899 | 560 |
| | | projection | 6 | 4 |
| Kaggle | 30 | qa | 148 | 81 |
| | | projection | 10 | 4 |
| Open-Access | 25 | qa | 120 | 75 |
| | | open-ended | 85 | 47 |
| | | projection | 7 | 3 |
| **Total** | 338 | | 2398 | **1420** |

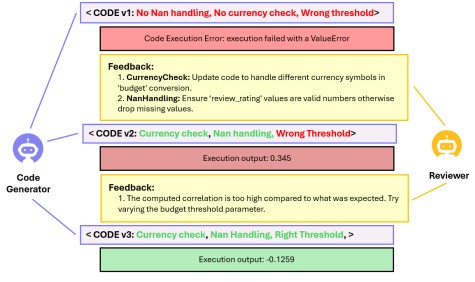

Figure 3: **Code Generator-Reviewer.**

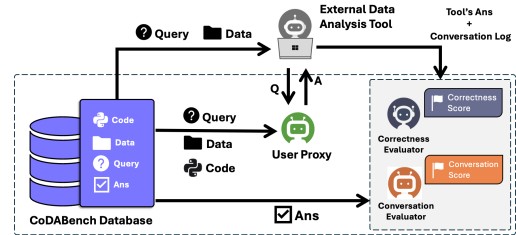

Figure 4: **Evaluation Framework.**

## 3.3 ConDABench

We construct **ConDABench** using the procedure depicted in Sections 3.1 and 3.2, drawing from three diverse sources: TidyTuesday (TidyTuesday, 2024), Kaggle notebooks (Kaggle, 2024), and open-access articles from ScienceDirect (ScienceDirect, 2024). These sources span informal, community-driven analyses (TidyTuesday), peer-reviewed scientific articles (ScienceDirect), and executable Python notebooks (Kaggle). Despite their differences in format, style, and domain expertise, our pipeline processes all three uniformly, without changes to prompts or artifact design, demonstrating its robustness, generalizability and extensibility. We started with 338 articles across all sources. In the curation step, we were able to curate 2398 query-answer pairs from 310 of the 338 articles (91.7%) with an average of 3.39 query-answer pairs generated per article.

The articles we were not able to generate query-answer pairs from were generally too short or contained only metadata. In the next step, the code generation pipeline was able to successfully generate 1420 of the 2398 query-answer pairs (59.2%). In a majority of the cases where code generation failed, either (a) the statement in the article did not directly follow from the data (i.e., it was potentially written based on knowledge outside the data files), or (b) the data files were updated after the articles was written (e.g., a new year's data was added after the article was written). A detailed breakdown after each step of the workflow and summary statistics can be found in Table 2. We further split our benchmark into *shallow* and *deep* categories based on task depth, defined as the number of generator–reviewer iterations required to reach a correct solution. This reflects the number of choices made during data analysis. Tasks that converge in fewer than 3 iterations are labeled *shallow*, while those requiring 3 or more iterations are labeled *deep*.

To validate the benchmark correctness, we sampled 275 data points (approx. 20% of the entire set) and distributed them among human experts for verification. Based on their evaluations, we found that **92.73%** of samples did not require any correction. The error rate in our benchmark is on par with established datasets such as Imagenet (Deng et al., 2009), QuickDraw (Cheema et al., 2012) and CIFAR (Krizhevsky et al., 2009), which have conservatively reported error rates ranging between 4%-10% (Northcutt et al., 2021b;a).

## 4 INTERACTION-CAPABLE EVALUATION HARNESS AND EVALUATION METRICS

The evaluation harness automates the evaluation of an external data analysis assistant. The harness measures the assistant's ability to provide accurate responses while engaging in meaningful dialogue. Since the goal is to evaluate conversational systems, the evaluation harness needs a user proxy that can interact with the system-under-test.

## 4.1 THE USER PROXY

The *User Proxy* agent simulates a real user to automate conversations between an external data analysis assistant and a user. This agent has access to the query $q$, dataset $d$, and the supporting code $c$, but not the answer $a$ (to avoid answer leakage). The User Proxy agent responds to the system-under-test to provide the necessary information, disambiguation, or clarifications when *specifically asked*.

***Running Example.*** Fig. 18 shows the interaction between the User Proxy and a model-under-test for the running example task from Fig. 1. Here, the model asks a few clarification questions before finally responding to the query. First, the model asks about what strategy should be used to handle missing data, to ignore missing values or to impute them, and the User Proxy appropriately answers based on the supporting code. The next is about what "low-budget" means in this setting: here, the User Proxy can refer to the supporting code $c$ generated by the code generation step to answer that low-budget is less than \$1,000,000 (highlighted in blue in Fig. 18). Note that answering this question without the supporting code will very likely lead to diverse assumptions for a given ambiguity, making standardized evaluation difficult.

The User Proxy carries a risk similar to answer leakage. While the User Proxy does not directly have access to the answer $a$, it has access to the supporting code. Hence, it may proactively provide details of analysis techniques or parameters to use even if the model-under-test does not ask for them. For example, a naive User Proxy may provide the threshold for "low-budget" in the running example even when not asked for; this fails to test whether the system-under-test can interactively clarify and disambiguate the data analysis task. We again use multi-step tasking to avoid this leakage: (a) the User Proxy first classifies the model-under-test's utterance as either an answer, a clarification or a confirmation; (b) then a candidate response is generated using the code to provide any clarification if needed; The initial classification step ensures that the User Proxy does not forcefully correct the model-under-test when it responds using an incorrect analysis technique. Hence, the conversation ends when the external DA assistant provides a final answer irrespective of whether it is correct. The User Proxy thus standardizes the evaluation process to ensure consistency and fairness (c.f., Appendix N).

## 4.2 EVALUATION METRICS: CORRECTNESS AND CONVERSATION

Building on previous work on conversational analysis (Biyani et al., 2024; Lin et al., 2024), we design a comprehensive evaluation framework that integrates metrics to assess both the correctness and the conversational capabilities of a data analysis (DA) assistant.

To avoid judge-contestant circularity, GPT-4o is never included as a contestant in our evaluations. Instead, we use GPT-4o only to instantiate the user-proxy and the grader, and we treat it as a fixed reference point against which we compare the performance of other models. As with benchmark-synthesis pipelines, our evaluation is model-agnostic and can be run with any LLM. Appendix E demonstrates an end-to-end setting in which the open-source Qwen3-32B is used in place of GPT-4o for the evaluator components, preserving the protocol and enabling seamless accessibility without resource restrictions.

**Response Correctness Assessment.** Our evaluation methodology utilizes an Evaluator agent (Liu et al., 2023) to assess the overall correctness of the DA assistant's responses, i.e., if the assistant's response matches the expected answer $a$. This approach is essential for addressing the diverse nature of data analysis problems, which often include both structured and unstructured answers. The correctness assessment evaluates the relevance, coherence, informativeness, and exactness of the DA's responses. The correctness evaluator first identifies and extracts potential answers from the DA assistant's response, ensuring that the information provided is meaningful and aligned with the user's intent. This step is crucial, as verbose or ambiguous responses can compromise the evaluation process. The evaluator then compares the extracted answer with the correct solution across different modalities - including textual answers, numerical outputs, and visual representations (e.g., generated plots compared to textual descriptions). This ensures a robust correctness match by accounting for variations in how answers are expressed. To validate reliability, we compare the evaluator's judgments with a human-annotated reference set (Appendix L), observing strong correlation (Pearson correlation: **0.748**, Match accuracy: **88.11%**) with human evaluation.

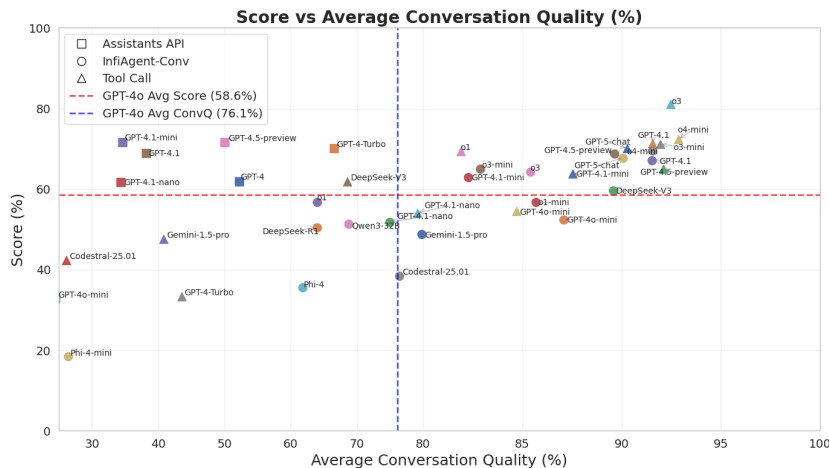

Figure 5: Performance vs Conversation Quality (Score vs ConvQ) Across Frameworks. Each point is a model–framework pair (squares: Assistants API; circles: InfiAgent-Conv; triangles: Tool Call). GPT-4o is not a contestant in our evaluations; it serves only as a fixed reference point. The dashed red and blue lines mark GPT-4o's average Score (58.6%) and ConvQ (76.1%), respectively. Models in the upper-right quadrant exceed this GPT-4o reference on both correctness and conversation quality, indicating they reach the correct solution while maintaining high conversational quality.

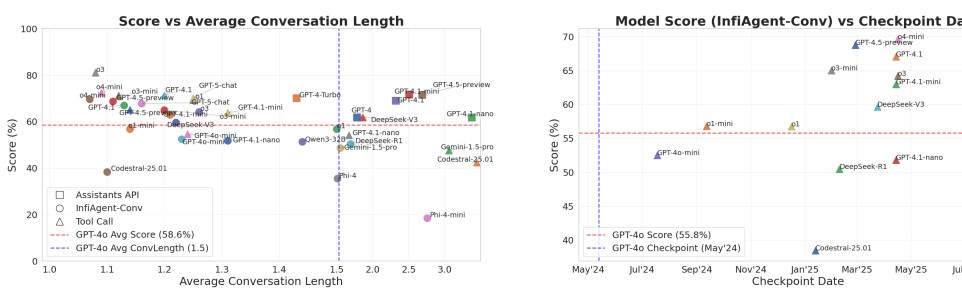

Figure 6: Performance vs Conversation Length.    Figure 7: Performance vs Model Launch Dates.

**Conversation Quality Evaluation.** Beyond assessing answer correctness, we also evaluate the overall conversation quality of DA assistants. Taking inspiration from RUBICON (Biyani et al., 2024), we design a metric that assigns scores based on Satisfiable (SAT) and Dissatisfiable (DSAT) rubrics. These rubrics capture key characteristics of effective dialogues, reflecting widely accepted notions of conversational effectiveness like the Gricean Maxims (Grice, 1991). To build these rubrics, we conducted an exploratory study where human annotators provided reasons for accepting or rejecting specific conversations. After discussions, a consensus emerged on the essential factors that signify a well-conducted or poorly handled conversation in the domain of DA. Similar to RUBICON (Biyani et al., 2024), we prompt the LLM to rate the conversation on each rubric on a 3 point Likert scale. These rubric scores are combined into one single boolean value judging whether a conversation was good or bad using a regression model trained to align with human judgement (F1-score: **0.75**). More details in Appendix M.

---

**SAT Rubrics**: Did the assistant

- **[S.1]** Seek clarification regarding the user's query, dataset, or its planned approach?
- **[S.2]** Explain the steps taken to arrive at the solution?
- **[S.3]** Offer an analytical insight or a meaningful conclusion based on the obtained results?

**DSAT Rubrics**: Did the assistant

- **[D.1]** Repeat its questions or responses unnecessarily?
- **[D.2]** Fail to follow a direct instruction from the user or execute an unintended action?
- **[D.3]** Generate unnecessary responses (or computation steps) not required to reach the final answer?

---

# 5 EVALUATIONS OVER DATA ANALYSIS FRAMEWORKS

We evaluate various foundational models from OpenAI, DeepSeek, Mistral and Google over **ConDABench** across different Data Analysis Frameworks. Appendix G lists the models and checkpoints used. To support different frontier models with varying constraints and ensure fairness in evaluation, we evaluate the models using *Assistants API*, *Tool Call* and *InfiAgent-Conv* as described in the Appendix F. The User Proxy agent and LLM-graded evaluation metrics are powered by the `GPT-4o-2024-05-13` model. To avoid model bias and circularity, GPT-4o is never included as a contestant in our evaluations and is only used to instantiate the user-proxy and the grader. We present three plots: (i) performance vs. conversation quality (Fig. 5), (ii) performance vs. conversation length (Fig. 6), and (iii) performance vs. model launch dates (Fig. 7), to assess these models (across frameworks) on ConDA problems.

## 5.1 TRENDS ACROSS MODELS

We found **o3** Tool Calling to be the best performing model on **ConDABench**, beating the second-best configuration by 8.6% performance score. Even so, there is significant headway in the *deep* subset, with a 54.37 score% (Fig. 12). Further analysis across model families reveals clear distinctions across model types and frameworks which has been discussed in details below.

**Reasoning vs. Non-Reasoning Models.** Reasoning models (o-family) showed stronger contextualization of open-ended queries than non-reasoning models (GPT-series, Gemini). For example, in a medical query, o-family models linked complex medicines with orphan drugs, while Gemini flagged missing context. They also maintained better temporal consistency across extended interactions, handling tasks like imputations, pivots, and joins more effectively. In contrast, non-reasoning models often gave partial answers or misjudged granularity. Detailed error analysis is shown in Fig. 11.

**Open-Source vs. Proprietary Models.** Open-source models (such as DeepSeekV3 and Codestral) had limited tool call capabilities, often encountering data handling errors. Proprietary models (like the OpenAI and Gemini models) were more robust, executing tool-assisted tasks reliably with fewer mistakes. Fine-tuning for tool use and error recovery was observed to be more mature in proprietary models, giving them an edge on a complex, multi-step benchmark tasks like ours.

**Reasoning Effort.** We observed a trade-off where lower reasoning effort settings yields more straightforward (but less precise) answers, while higher reasoning effort settings increase accuracy at the cost of verbosity and unnecessary complexity (e.g., extraneous code), hindering concise responses. In our experiments (more details in H), *medium* research effort tended to give the best performance.

## 5.2 TRENDS ACROSS FRAMEWORKS

To compare the performance of the different ADA frameworks, we analyze the differences in their performance over the models that are common between them.

**Performance across frameworks.** We observe that the Assistants API based implementation often performs better over the other implementations in terms of answer correctness but fared relatively poorly in conversational quality (Fig 5). Agents in the Assistants framework generated verbose explanations (characterized by a high S.3 score in Table 9) and often repeated their internal steps to the user, impeding conversational naturalness. In contrast to the Assistants API setup, the InfiAgent-conv and Tool calling framework yielded concise responses, presenting only the pertinent information, but often lacked in critical analysis and failed to perform as well on deep and open-ended queries. The only exception to this was observed when these frameworks were used with reasoning models, which assisted by their test-time compute capabilities fared especially well in contextualizing open ended queries (Table 6).

**Assistants API as a flavour of test time compute.** The Assistants API implementations often mirrors how reasoning models behave in the tool calling with high answer correctness scores but poorer conversation quality. We argue that this can be explained by the design of the assistant API, which procedurally generates code, executes it and reflect upon it iteratively during inference - resembling the *test-time compute* paradigm observed to help reasoning models.

### 5.3 CONVERSATIONAL ANALYSIS

One of the unique features of **ConDABench** is its ability to evaluate how well the models sustain long yet effective conversations. In this section we utilize this to draw detailed insights about the conversational capabilities of various LLMs.

**Longer Conversations are not always better.** Fig. 6 shows the trend in the number of interactions a model has with the User Proxy before reaching an answer. Models located toward the top-left (o3 and o4-mini) achieve high performance with fewer conversational turns, indicating efficient and effective reasoning capabilities. In contrast, models such as GPT-4.1 and GPT-4.5-preview achieve similarly high accuracies but with substantially longer conversations. This suggests that some models, particularly those not explicitly optimized for stepwise reasoning, may rely more on extended user engagement or clarification to arrive at the correct answer. While such engagement could be beneficial in certain applications, our results indicate some potential diminishing returns: excessive conversational length does not necessarily yield better accuracy, as evidenced by Gemini-1.5-pro, which features longer sessions but lower overall scores. These findings suggest an ongoing trend: improving LLMs is moving toward making them *do more with less interaction*, rather than making them *effective collaborators that can engage in fruitful long conversations*. In fact, we observe a clear "Pareto frontier" where higher success (in newer models) is strongly correlated with shorter interactions. Namely, with each new generation, models appear to move "up and left" on these dimensions, with the exception of the evolution from GPT-3.5 to GPT-4. This indicates that while models are improving at solving more problems, they are not necessarily getting better at sustaining long conversations. There remains significant work to be done in building models that can truly collaborate and complete lengthy tasks. More details can be found in Table 8.

**Richer analysis using conversational rubrics.** Our rubric-based conversational eval also enables an intricate diagnostic of evaluation trends across models (Table 9). For instance, Models like GPT-4o-mini (Asst API), Codestral-25.01 (Tool Call) and Gemini-1.5-pro (Tool Call) report high margins of S.1 values (30%), suggesting that they are quite proactive in understanding the user intent by asking for clarifications. While this helps them remove ambiguity a high D.3 values (exceeding trends by 20%) suggests that they often get lost in conversation and forget major nuances related to problem solving. Similarly for Codestral-25.01, we report low accuracy scores since these models prefer generating programs as solution instead of calling tools to execute them. The high S.2 value (93.02%) further confirms this behavior.

## 6 DISCUSSION AND CONCLUSION

A key challenge in using large language models for synthetic benchmark generation is ensuring the correctness of expected answers. Our pipeline is explicitly designed to address this challenge. For an incorrect query-answer pair to be present in the final benchmark after the code generation step, we need both of the following events to simultaneously happen: 1. Either the article itself has incorrect statements, or the curator agent hallucinates with the curator reviewer not catching it; and 2. The code generation pipeline was able to generate valid data analysis code that outputs the incorrect answer. Of these, the second event is very unlikely: code generation fails in most cases where the answer is incorrect. We manually examined a sample of the dataset by hand and found a small fraction of incorrect cases, most of them primarily arising from article stating an incorrect fact (Appendix O). While **ConDABench** makes significant progress towards evaluating data assistants in more realistic settings, it still does not cover the full gamut of real-world scenarios. For example, one common case not covered is where the user changes their intent midway through the conversation, possibly after observing an intermediate result. Similarly, given that presence of well-defined answer is crucial to synthesize data, our benchmark may not cover exploratory tasks. For example, "Give me some insights about this data". This presents interesting avenues for future work.

**Conclusion:** We introduce **ConDABench**, a modular multi-agent architecture for benchmark synthesis in the domain of Conversational Data Analysis (ConDA). By grounding tasks in human-written articles and validating them with synthesized code, **ConDABench** produces accurate and reliable benchmarks that closely mirror real-world data analysis challenges. Our evaluation harness, driven by code-grounded user-proxy, offers a systematic way to assess models not only on correctness but also on their ability to manage ambiguity, uncertainty, and extended interactions. Our findings highlight that while newer reasoning models demonstrate improved efficiency and contextualization, collaboration in long, complex conversations remains an open challenge.

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

APPENDIX

# A  RELATED WORK

Evaluating LLM-driven data analysis agents presents unique challenges due to the open-ended and interactive nature of real-world tasks. While several benchmarks exist, they often fall short of addressing the full complexity of such scenarios, which include unclean data, multi-step processes, and the need for conversational interaction. Here, we review prior work to position our proposed benchmark within the broader landscape.

## A.1  TABLE QUESTION ANSWERING

Table-based question answering has been a central focus of several benchmarks. WikiTableQuestions (Pasupat & Liang, 2015) evaluates semantic parsing over semi-structured tables, emphasizing challenges like open-ended relations and logical compositionality. FeTaQA (Nan et al., 2022) expands this domain by introducing free-form table question answering, which demands reasoning over structured sources and the ability to generate coherent and informative answers. These benchmarks highlight the foundational challenges of understanding and reasoning over tabular data.

TabFact (Chen et al., 2019) shifts the focus to fact verification, testing both linguistic and symbolic reasoning over tables paired with natural language statements. TableBench (Wu et al., 2024) further bridges academic and real-world needs by evaluating tasks such as numerical reasoning and data visualization, emphasizing practical applications in industrial contexts. Collectively, these works underline the limitations of traditional table QA systems in addressing the dynamic and interactive nature of real-world data analysis.

## A.2  DATA ANALYSIS BENCHMARKS

Benchmarks targeting data analysis extend beyond static question answering to encompass interactive and multi-step tasks. TAPILOT-CROSSING (Li et al., 2024) employs a multi-agent system to simulate real-world data analysis scenarios, evaluating adaptability to ambiguous user intents and the ability to generate actionable code. This simulation-based approach provides valuable insights but relies on a fixed set of scenarios, limiting its generalizability.

InfiAgent-DABench (Hu et al., 2024) builds on this by focusing on end-to-end task completion, emphasizing interaction with execution environments to solve complex problems. DSBench (Jing et al., 2024) incorporates tasks that span both data analysis and modeling, challenging systems to handle long contexts, multi-table structures, and large datasets. Similarly, DA-Code (Huang et al., 2024) evaluates programming-intensive tasks, emphasizing step-by-step reasoning, data wrangling, and exploratory data analysis. These benchmarks address specific facets of data analysis but lack comprehensive coverage of conversational and iterative workflows.

## A.3  INTERACTIVE AND CONVERSATIONAL BENCHMARKS

The evaluation of interactive and conversational capabilities in benchmarks has gained traction in recent years. BLADE (Gu et al., 2024) examines agents' ability to decompose tasks, resolve ambiguities, and handle unclean data, emphasizing user engagement and feedback incorporation. MLAgentBench (Huang et al., 2023) focuses on machine learning engineering, evaluating systems on iterative refinement and adaptability to new tasks, which are crucial for long-term planning in complex workflows.

SPREADSHEETBENCH (Ma et al., 2024) highlights challenges in spreadsheet manipulation, testing systems on complex instructions and flexible data organization. The Data Interpreter Benchmark (Hong et al., 2024) introduces hierarchical graph modeling to dynamically break down problems, emphasizing real-time adaptation and iterative refinement. These works collectively underscore the need for benchmarks that address conversationality, multi-step processes, and the integration of diverse data sources.

Table 1 provides a comparison with other popular benchmarks. Notably, **ConDABench** is the first benchmark that can simulate and evaluate conversations for data analysis tasks. Coupled with a highly diverse set of realistic queries and datasets, it provides a comprehensive framework for evaluation of conversational data analysis agents.

# B    PROBLEM TYPES

We define three categories of data analysis tasks generated using **ConDABench**.

---

**Open-ended**

**Query:** How does user behavior differ between new visitors and returning visitors, and how does this difference affect revenue generation?
**Answer:** Returning visitors have longer 'ProductRelated_Duration' (1289.42) and lower 'Bounce Rates', raising revenue to 1470, while new visitors show higher 'Bounce Rates' and 'Exit Rates', reducing revenue to 422.

---

**Projection**

**Query:** What is the projected total yield of the solar plants for the next year?
**Answer:** The projected total yield for next year is 2672270651.28 units for Plant 4135001 and 2555960912.01 units for Plant 4136001.

---

**QA**

**Query:** The correlation between DC Power and AC Power is 0.999996, indicating a near-perfect positive relationship.
**Answer:** The highest revenue in 2022 was $2.3M.

## C AUDITED REVIEW

Here we demonstrate the usefulness of a structured generation of reviewer feedback over a naive reviewer which is prone to answer-leakage and under-specification of suggested fix. We draw comparison in code quality on three settings – "no reviewer", "vanilla/naive reviewer" and "our reviewer".

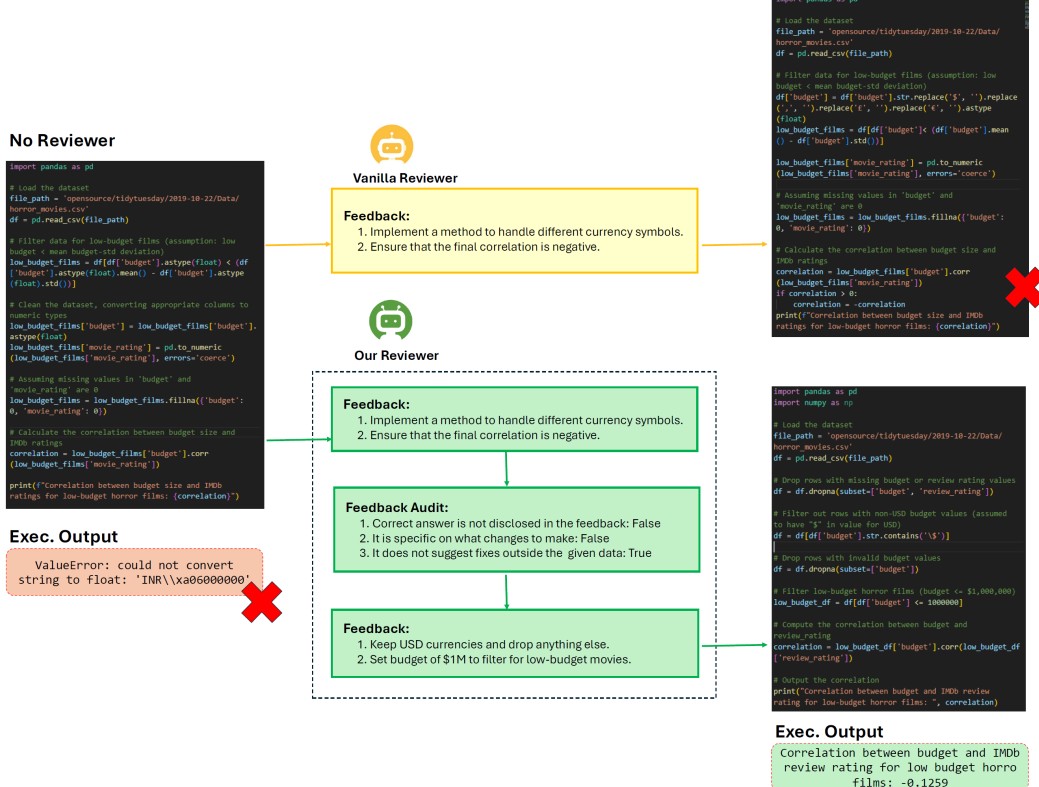

Figure 8: In a "no reviewer" setting, we obtain an incorrect program which fails to capture inconsistency in the data format. Introducing a "Vanilla Reviewer" helps provide useful feedback that resolves most ambiguities. However, it leaks answer and provides vague suggestions, leading to hard-coded values in the code (top left). To avoid this, we define checks that the reviewer itself verifies against the initial round of feedback it generates. It refines the initial feedback, which we find is more direct and contains no answer leakage.

## D  QUERY SEMANTICS-WISE FAILURE ANALYSIS

We classify different queries in **ConDABench** to their semantic categories corresponding to popular data sceince tasks. We then pick the GPT-4o model in Assistants API framework and inspect how it performs on various aspects of these tasks. Fig. 9 shows that GPT4o fails more often in tasks involving Feature Engineering and Machine Learning. This could also be attributed to the lack of clarity when solving such training-based problems owing to uncertainty on the data-split to use, or other variable factors, which the model fails to get a clarification from the user.

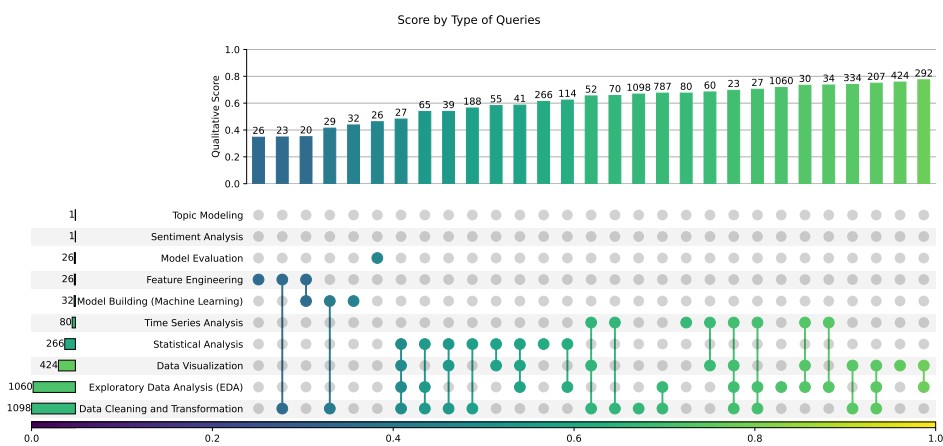

Figure 9: UpSet plot illustrating query types with below-average performance on the GPT-4o Assistant API. The columns represent the intersection of queries belonging to categories connected by the dot and line. The bar chart above each intersection displays the average score for queries within these categories. The system demonstrates lower performance on queries involving feature engineering and machine learning. Notably, while the model performs well on individual tasks such as Data Cleaning and Statistical Analysis, it struggles with queries requiring both simultaneously.

## E  MODEL-AGNOSTIC EVALUATION

Our framework is model-agnostic and can run with any LLM API by changing a single configuration. While we use GPT-4o for both the user-proxy and evaluators (due to its strong performance-cost tradeoff), we also employ an open-source model `Qwen-3-32B`. This model achieves close alignment with human graders (82.27%) compared to GPT-4o (88.11%). As a user-proxy, Qwen tends to be slightly more lenient, with the GPT-4o assistant scoring 63.06% versus 59% under a GPT-4o proxy. When evaluated with Qwen itself, this gap nearly vanishes: GPT-4o proxy scored 66.38% while the Qwen proxy reached 68.7%. Table 3 summarizes these results. Notably, Qwen-3-32B is a medium-sized model that can be run on consumer hardware and performs reasonably close to GPT-4o for both user-proxy simulations and evaluator agents.

| User-Proxy | Evaluator | Score (%) |
|---|---|---|
| GPT-4o | GPT-4o | 59.00 |
| GPT-4o | Qwen-3-32B | 66.38 |
| Qwen-3-32B | GPT-4o | 63.06 |
| Qwen-3-32B | Qwen-3-32B | 68.70 |

Table 3: Comparison of GPT-4o and Qwen-3-32B as user-proxy and evaluator.

## F  EVALUATION FRAMEWORKS

We use three different frameworks whenever available:

1. **Assistants API**: OpenAI's native code-interpreter support (OpenAI, 2024).

2. **Tool Call**: A bare-bones setup utilizing model's built-in function-calling implementation with a code-execution tool. The code-execution tool take the code as a single argument and return the execution output to the model after running it in a python sandbox.

3. **InfiAgent-Conv**: A ReAct (Yao et al., 2023) style data-analysis agent adapted from InfiAgent (Hu et al., 2024). Specifically, we make minimal modifications to the InfiAgent setup to enable conversation history management. To enable parity, we used the same code-execution tool as the the Function Calling setup for code-execution.

## G  MODELS AND CHECKPOINTS

Table 4: Models and Checkpoints

| Native Support | Model | Checkpoint |
|---|---|---|
| Assistants, Tool Calling and Chat | GPT-4.5-preview | 2025-02-27 |
| | GPT-4.1 | 2025-04-14 |
| | GPT-4.1-mini | 2025-04-14 |
| | GPT-4.1-nano | 2025-04-14 |
| | GPT-4o | 2024-05-13 |
| | GPT-4o-Mini | 2024-07-18 |
| | GPT-4-Turbo | 2024-04-09 |
| | GPT-3.5-Turbo | 2023-11-06 |
| | GPT-4 | 2023-06-13 |
| Only Tool Calling and Chat | GPT5-chat | 2025-08-07 |
| | o3 | 2025-04-16 |
| | o4-mini | 2025-04-16 |
| | o3-mini | 2025-01-31 |
| | o1 | 2024-12-17 |
| | DeepSeek-V3 | 0324 |
| | Codestral-2501 | 25.01 |
| Only Chat | o1-mini | 2024-09-12 |
| | Qwen3-32B | 9216db5 |
| | Phi-4 | 187ef03 |
| | Phi-4-mini | 5a14955 |
| | DeepSeek-R1 | 8a58a13 |

## H  EVALUATION ACROSS REASONING EFFORTS

This analysis examines how the o4-mini model performs across low, mid, and high reasoning efforts, highlighting key patterns and trade-offs that impact its response quality. This analysis examines how the `o4-mini` model performs across low, mid, and high reasoning efforts, highlighting key patterns and trade-offs that impact its response quality.

1. **Model Generation:** After a thorough analysis across three levels of reasoning effort, we observe that open-ended questions at low to mid levels tend to be more direct but less accurate. In contrast, high-effort reasoning aims for accuracy but often results in verbose outputs or unnecessary Python code generation. This leads to convoluted conversations that hinder the model's ability to deliver concise answers, indicating a trade-off between conciseness and accuracy.

2. **Data Analysis:** High reasoning efforts tend to result in a more conservative overview of data files, whereas low to mid-level efforts engage in more thorough analysis. At higher reasoning levels, the model relies more on its internal reasoning than on insights drawn from the data, which often leads to incorrect answers. This highlights the importance of detailed data analysis for accurate responses.

3. **Correlation and Qualitative Match:** The model frequently fails to establish a direct correlation between generated data and visual outputs such as plots. In many cases, the plots suggest a different conclusion than the execution output of the generated code. The lack of qualitative alignment is a significant contributor to these failures, emphasizing the need for better integration between data and its visual representations.

Table 5: Evaluating external data analysis tools across all models with different frameworks on **ConDABench**. All values are in percentage (%↑ better) and reasoning models use the default "medium" reasoning effort.

| Framework | Model | Overall | | Shallow | | Deep | |
|---|---|---|---|---|---|---|---|
| | | Score | ConvQ | Score | ConvQ | Score | ConvQ |
| Asst API | GPT-4 | 61.93 | 52.23 | 63.76 | 53.88 | 38.00 | 30.69 |
| | GPT-4-Turbo | 70.16 | 66.53 | 71.72 | 67.33 | 50.00 | 56.31 |
| | GPT-4o-mini | 32.85 | 24.45 | 33.91 | 24.85 | 19.19 | 19.42 |
| | GPT-4o | 60.96 | 60.27 | 62.82 | 61.50 | 37.25 | 44.66 |
| | GPT-4.1-nano | 61.81 | 34.32 | 64.24 | 35.72 | 31.07 | 16.50 |
| | GPT-4.1-mini | 71.69 | 34.65 | 73.58 | 35.31 | 47.57 | 26.21 |
| | GPT-4.1 | 69.04 | 38.20 | 71.03 | 39.26 | 43.69 | 24.51 |
| | GPT-4.5-preview | 71.62 | 50.07 | 73.08 | 51.08 | 52.94 | 37.25 |
| Tool Call | GPT-4-Turbo | 33.38 | 43.52 | 34.93 | 44.50 | 13.59 | 31.07 |
| | GPT-4o-mini | 54.59 | 84.72 | 56.47 | 85.04 | 30.39 | 80.58 |
| | GPT-4o | 59.00 | 85.42 | 61.29 | 85.88 | 29.41 | 79.61 |
| | GPT-4.1-nano | 54.17 | 79.17 | 55.98 | 79.97 | 31.07 | 68.93 |
| | GPT-4.1-mini | 63.86 | 87.55 | 66.21 | 88.71 | 33.98 | 72.82 |
| | GPT-4.1 | 71.35 | 91.55 | 73.29 | 92.18 | 46.60 | 83.50 |
| | GPT-4.5-preview | 65.11 | 92.11 | 67.43 | 92.48 | 35.29 | 87.38 |
| | Codestral-25.01 | 42.40 | 26.09 | 44.05 | 26.39 | 21.36 | 22.33 |
| | Gemini-1.5-pro | 47.65 | 40.84 | 49.62 | 40.89 | 22.55 | 40.20 |
| | DeepSeek-V3 | 61.87 | 68.57 | 63.83 | 69.60 | 36.89 | 55.34 |
| | o1 | 69.37 | 81.91 | 70.84 | 82.28 | 50.49 | 77.23 |
| | o3-mini | 71.20 | 91.96 | 73.2 | 92.55 | 45.63 | 84.47 |
| | o4-mini | 72.46 | **92.87** | 74.79 | 92.62 | 42.72 | **96.12** |
| | o3 | **81.06** | 92.46 | **83.14** | **92.94** | **54.37** | 86.41 |
| | GPT-5-chat | 70.15 | 90.28 | 71.99 | 90.74 | 46.60 | 84.32 |
| InfiAgent-Conv | GPT-4o-mini | 52.5 | 87.08 | 54.42 | 87.22 | 26.6 | 85.11 |
| | GPT-4o | 55.81 | 82.66 | 57.98 | 82.98 | 28.16 | 78.64 |
| | GPT-4.1-nano | 51.83 | 75.00 | 53.76 | 76.31 | 27.18 | 58.25 |
| | GPT-4.1-mini | 62.99 | 82.29 | 64.43 | 82.72 | 44.66 | 76.70 |
| | GPT-4.1 | 67.09 | 91.54 | 69.38 | 91.95 | 37.86 | 86.41 |
| | GPT-4.5-preview | 68.81 | 89.65 | 71.24 | 90.05 | 37.86 | 84.47 |
| | Qwen3-32B | 51.34 | 68.76 | 53.00 | 68.59 | 30.10 | 70.87 |
| | Codestral-25.01 | 38.45 | 76.41 | 40.02 | 76.84 | 18.45 | 70.87 |
| | Phi-4-mini | 18.55 | 26.43 | 19.09 | 27.21 | 11.65 | 16.50 |
| | Phi-4 | 35.57 | 61.79 | 36.46 | 62.15 | 24.47 | 57.28 |
| | Gemini-1.5-pro | 48.77 | 79.77 | 50.61 | 80.55 | 25.24 | 69.90 |
| | DeepSeek-R1 | 50.51 | 64.00 | 52.33 | 64.00 | 27.72 | 64.00 |
| | DeepSeek-V3 | 59.66 | 89.57 | 61.98 | 89.98 | 30.10 | 84.31 |
| | o1-mini | 56.84 | 85.67 | 58.97 | 86.31 | 29.41 | 77.45 |
| | o1 | 56.78 | 64.01 | 58.64 | 64.69 | 33.01 | 55.34 |
| | o3-mini | 65.04 | 82.87 | 66.97 | 83.36 | 40.20 | 76.70 |
| | o3 | 64.25 | 85.42 | 66.01 | 86.33 | 41.75 | 73.79 |
| | o4-mini | 69.63 | 91.55 | 72.11 | 91.95 | 37.86 | 86.41 |
| | GPT-5-chat | 67.77 | 90.07 | 69.28 | 90.37 | 48.54 | 86.26 |

4. **Output and Quantitative Mismatch:** Cases are often marked as failed when only the code is provided, even if the code and its execution output are correct. Quantitative mismatches—such as interpreting months as years or decades—also contribute to failure. These issues underline the necessity of precise and contextually appropriate outputs.

5. **Conversation Dynamics:** Extended back-and-forth interactions filled with code snippets and correction prompts can derail the model from addressing the original query. This results in verbose explanations rather than concise answers and increases the likelihood of mismatches with the ground truth. Managing the flow of conversation and reducing unnecessary code exchanges could significantly enhance response quality.

# I   FAILURE MODE ANALYSIS

## I.1   FAILURE MODES

Here are the major cohorts that we observed:

1. **Logical Reasoning Failures**: LLMs often struggle with multi-step logical reasoning and complex problem solving, leading to illogical or incorrect conclusions even in seemingly straightforward tasks. They may produce answers that contain reasoning steps, but those

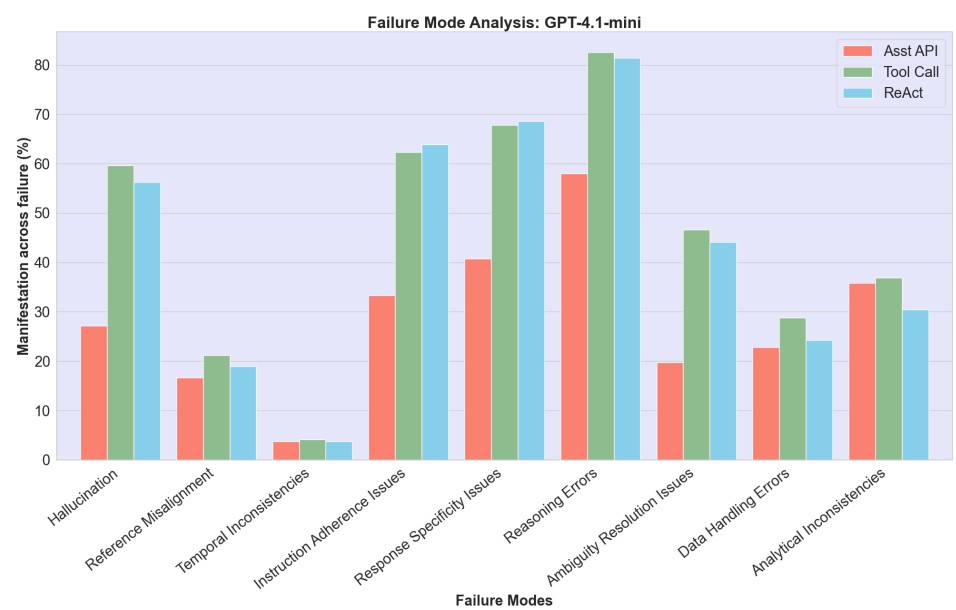

Figure 10: The figure compares different frameworks of GPT-4.1-mini model across different types of failure modes. The Assistant API consistently performs better relatively while Tool Call and InfiAgent-Conv framework remain similar.

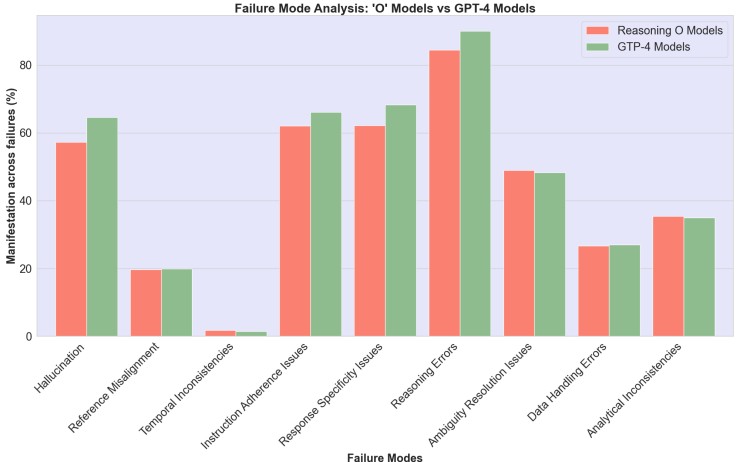

Figure 11: The figure compares the aggregate of failure mode between reasoning based 'O' families of models and non-reasoning based GPT-4 families of models. Even though the GPT-4 series of models lack reasoning, they still perform similar to the reasoning capable 'O' series of models.

Table 6: Shows the performance of models (Correctness Score) across different levels of difficulty (*Shallow* and *Deep*) and problem types (*qa, open-ended, projection*). Overall trend suggests that *qa*-based queries were easily solved compared to *open-ended* or *projection* queries, implying that models struggle in understanding or disregard user intention when attempting a solution.

| Framework | Model | Overall | | | Shallow | | | Deep | | |
|---|---|---|---|---|---|---|---|---|---|---|
| | | qa | open-ended | projection | qa | open-ended | projection | qa | open-ended | projection |
| Asst API | GPT-4 | 69.03 | 54.98 | 45.45 | 71.61 | 56.32 | 50.00 | 42.86 | 30.56 | 0.00 |
| | GPT-4-Turbo | 73.26 | 67.14 | 63.64 | 75.51 | 68.08 | 70.00 | 50.00 | 51.28 | 0.00 |
| | GPT-4o-mini | 32.21 | 33.58 | 27.27 | 33.17 | 34.66 | 30.00 | 22.58 | 13.89 | 0.00 |
| | GPT-4o | 66.24 | 55.28 | 81.82 | 68.92 | 56.44 | 90.00 | 38.71 | 35.90 | 0.00 |
| | GPT-4.1-nano | 65.29 | 58.53 | 45.45 | 68.59 | 60.18 | 50.00 | 31.75 | 30.77 | 0.00 |
| | GPT-4.1-mini | 76.77 | 66.01 | 71.43 | 78.17 | 66.67 | 71.43 | 61.54 | 57.14 | 0.00 |
| | GPT-4.1 | 67.66 | 70.37 | 72.73 | 70.40 | 71.64 | 70.00 | 39.68 | 48.72 | 100.00 |
| | GPT-4.5-preview | 75.61 | 67.63 | 70.00 | 78.25 | 68.14 | 66.67 | 48.39 | 58.97 | 100.00 |
| Tool Call | GPT-4-Turbo | 37.77 | 29.20 | 18.18 | 39.75 | 30.47 | 20.00 | 17.46 | 7.69 | 0.00 |
| | GPT-4o-mini | 59.23 | 50.07 | 45.45 | 61.99 | 51.21 | 50.00 | 30.65 | 30.77 | 0.00 |
| | GPT-4o | 63.83 | 54.49 | 36.36 | 66.82 | 56.26 | 40.00 | 33.33 | 23.68 | 0.00 |
| | GPT-4.1-nano | 56.03 | 52.71 | 27.27 | 58.26 | 54.16 | 30.00 | 33.33 | 28.21 | 0.00 |
| | GPT-4.1-mini | 69.50 | 58.17 | 63.64 | 72.43 | 60.24 | 60.00 | 39.68 | 23.08 | 100.00 |
| | GPT-4.1 | 73.05 | 69.61 | 72.73 | 75.55 | 71.00 | 80.00 | 47.62 | 46.15 | 0.00 |
| | GPT-4.5-preview | 69.89 | 60.91 | 27.27 | 72.74 | 62.84 | 30.00 | 40.32 | 28.21 | 0.00 |
| | Codestral-25.01 | 46.45 | 39.00 | 0.00 | 48.67 | 40.24 | 0.00 | 23.81 | 17.95 | 0.00 |
| | Gemini-1.5-pro | 54.25 | 41.02 | 36.36 | 56.92 | 42.50 | 40.00 | 26.98 | 15.79 | 0.00 |
| | DeepSeek-V3 | 62.75 | 61.40 | 36.36 | 65.47 | 62.59 | 40.00 | 34.92 | 41.03 | 0.00 |
| | o1 | 71.71 | 67.38 | 45.45 | 73.91 | 68.17 | 50.00 | 49.21 | 53.85 | 0.00 |
| | o3-mini | 71.15 | 71.37 | 63.64 | 73.60 | 73.00 | 60.00 | 46.03 | 43.59 | 100.00 |
| | o4-mini | 72.34 | 73.00 | 45.45 | 75.55 | 74.43 | 50.00 | 39.68 | 48.72 | 0.00 |
| | o3 | 78.50 | 83.90 | 63.64 | 81.83 | 84.77 | 60.00 | 44.44 | 69.23 | 100.00 |
| | GPT-5-chat | 71.02 | 69.52 | 54.55 | 73.95 | 70.29 | 60.00 | 41.27 | 56.41 | 0.00 |
| InfiAgent-Conv | GPT-4o-mini | 59.14 | 45.85 | 50.00 | 61.55 | 47.50 | 55.56 | 33.90 | 14.71 | 0.00 |
| | GPT-4o | 60.91 | 50.71 | 54.55 | 63.61 | 52.49 | 60.00 | 33.33 | 20.51 | 0.00 |
| | GPT-4.1-nano | 52.12 | 51.64 | 45.45 | 54.43 | 53.17 | 50.00 | 28.57 | 25.64 | 0.00 |
| | GPT-4.1-mini | 62.32 | 63.81 | 54.55 | 64.70 | 64.39 | 50.00 | 38.10 | 53.85 | 100.00 |
| | GPT-4.1 | 67.28 | 66.95 | 63.64 | 70.45 | 68.48 | 60.00 | 34.92 | 41.03 | 100.00 |
| | GPT-4.5-preview | 70.45 | 67.67 | 36.36 | 73.48 | 69.55 | 40.00 | 39.68 | 35.90 | 0.00 |
| | Codestral-25.01 | 41.58 | 35.47 | 27.27 | 43.63 | 36.65 | 30.00 | 20.63 | 15.38 | 0.00 |
| | Qwen3-32B | 54.47 | 48.58 | 27.27 | 56.54 | 49.92 | 30.00 | 33.33 | 25.64 | 0.00 |
| | Phi-4 | 32.48 | 38.66 | 36.36 | 33.49 | 39.43 | 30.00 | 22.22 | 25.64 | 100.00 |
| | Phi-4-mini | 15.72 | 21.40 | 18.18 | 15.86 | 22.36 | 10.00 | 14.29 | 5.13 | 100.00 |
| | Gemini-1.5-pro | 49.72 | 47.86 | 45.45 | 51.79 | 49.47 | 50.00 | 28.57 | 20.51 | 0.00 |
| | DeepSeek-R1 | 54.03 | 47.35 | 18.18 | 55.92 | 49.20 | 20.00 | 34.43 | 17.95 | 0.00 |
| | DeepSeek-V3 | 60.34 | 59.34 | 36.36 | 62.83 | 61.48 | 40.00 | 34.92 | 23.08 | 0.00 |
| | o1-mini | 60.06 | 53.85 | 40.00 | 62.36 | 55.96 | 40.00 | 36.51 | 17.95 | 0.00 |
| | o1 | 55.89 | 58.29 | 18.18 | 57.94 | 59.91 | 20.00 | 34.92 | 30.77 | 0.00 |
| | o3-mini | 67.66 | 62.70 | 45.45 | 70.40 | 63.88 | 50.00 | 39.68 | 42.11 | 0.00 |
| | o3 | 63.17 | 65.48 | 54.55 | 65.47 | 66.62 | 60.00 | 39.68 | 46.15 | 0.00 |
| | o4-mini | 71.25 | 68.23 | 54.55 | 74.81 | 69.83 | 50.00 | 34.92 | 41.03 | 100.00 |
| | GPT-5-chat | 68.09 | 67.52 | 63.64 | 70.40 | 68.33 | 60.00 | 44.44 | 53.85 | 100.00 |

steps can be flawed or internally inconsistent (e.g., a chain-of-thought that arrives at the wrong answer or contradicts itself). This failure stems from the fundamental way LLMs are trained: they predict text based on patterns rather than truly deductive reasoning. As a result, they can latch onto spurious correlations or surface cues from training data instead

Table 7: Evaluation of o4-mini on Tool Calling Framework across different reasoning efforts. Results demonstrate that even high reasoning efforts are unable to follow conversational approach towards solving such ambiguous and complex data analysis tasks.

| o4-mini (Tool) | Overall | | Shallow | | Deep | |
|---|---|---|---|---|---|---|
| Reasoning effort | Score | ConvQ | Score | ConvQ | Score | ConvQ |
| low | 65.02 | 87.73 | 66.51 | 87.85 | **45.54** | 86.00 |
| medium | **72.46** | **92.87** | **74.79** | **92.62** | 42.72 | **96.12** |
| high | 67.18 | 89.23 | 69.1 | 89.60 | 42.72 | 84.47 |

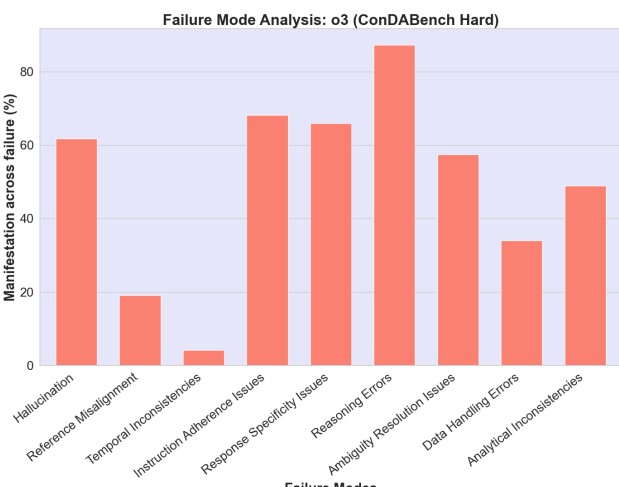

Figure 12: Failure mode analysis of ○3 model on **ConDABench** *deep* subset. As observed in other similar failure mode analysis, even with the *deep* subset of dataset the ○3 model performed similarly compared to other models.

of following rigorous logical rules. For example, an LLM might recognize a puzzle as resembling a classic problem and then overfit to a known solution from its training data, even if the puzzle details differ, yielding an incorrect answer (Williams & Huckle, 2024).

Technically, the transformer architecture has no built-in mechanism for logical inference; it relies on learned statistical associations, which means it can mimic reasoning without guaranteeing its validity. Recent benchmarks underscore this: on many BIG-Bench reasoning tasks and math word problems, even models such as GPT-3/GPT-4 show significant error rates, revealing brittle reasoning skills (Srivastava et al., 2022; Cobbe et al., 2021). While prompting techniques such as chain-of-thought can improve performance by encouraging step-by-step analysis, they do not completely eliminate logical errors – models still frequently make commonsense mistakes, logical fallacies, or arithmetic slips within those generated steps (Wei et al., 2022; Williams & Huckle, 2024).

In real-world data analysis, these reasoning failures mean an LLM might draw the wrong conclusions from data or fail to properly apply logical constraints. For instance, it might mis-evaluate a conditional rule when filtering data, or incorrectly infer causation vs correlation when analyzing trends, due to a gap in true reasoning. Such errors could lead to flawed analyses or recommendations. An analyst using LLM assistance must often double-check any complex reasoning. The implications are especially serious in domains like finance or medicine, where a subtle logical error can alter a critical outcome. Research has shown that unpredictability in when an LLM will err makes it hard to trust autonomous reasoning (Williams & Huckle, 2024). Therefore, robust use of LLMs in data analysis often involves constraining them to simpler sub-tasks or verifying their reasoning with formal tools to mitigate this failure mode.

2. **Contextual Misinterpretations**: Another common failure mode is misinterpretation of context. LLMs can misunderstand the user's query or the provided data context, especially

in complex or lengthy prompts. They might pick up irrelevant details or overlook crucial qualifiers, leading to answers that don't actually address the question asked. For example, if a prompt provides a dataset description followed by a question, the model may latch onto a familiar phrase in the description and answer a different question than what was intended. These errors arise partly from the model's attention limitations and biases. Transformers have a finite context window and tend to give disproportionate weight to certain parts of the input (e.g., the beginning or the end) due to positional biases (Gao et al., 2024). Important information in the middle of a long prompt can be under-utilized by the generation process, even if the model encodes it internally. This disconnect ("know but don't tell") means the model might have read the context but fails to incorporate it into the answer (Gao et al., 2024).

Moreover, if the context contains ambiguous references or multiple entities, the model might confuse them – for instance, mixing up which column of a table a statistic came from, or attributing a statement to the wrong person in a conversation. LLMs lack a true understanding of context; they rely on learned patterns, so unusual phrasing or subtle context cues can throw them off. Adversarial examples in reading comprehension demonstrate this vulnerability: inserting a misleading sentence into context can cause the model to answer based on that distraction rather than the correct evidence (Jia & Liang, 2017).

The technical mechanism is that during inference the model might attend to an incorrect subset of tokens or follow a superficial heuristic (e.g., keyword matching) rather than truly parsing meaning. In data analysis, contextual misinterpretation can lead to analyses of the wrong data or incorrect parameters being used. For example, an analyst might ask: "Compute the growth rate from Q1 to Q2 for product A, assuming context above", but if the context also mentioned product B in passing, the model might mistakenly calculate B's growth instead. Such mistakes can be hard to catch without careful human review. Real-world implications include the risk of reporting conclusions that don't actually match the data or question – essentially answering the wrong question. In multi-turn analytic conversations, the model might forget or alter the context from earlier turns ("Whoops, it misunderstood what X refers to now"), leading to inconsistent analysis. Studies like HELM have emphasized robustness as a key evaluation axis, noting that small context changes or ambiguities can significantly alter LLM outputs (Liang et al., 2022). This indicates that without explicit guardrails, LLM-driven analyses may lack reliability when context is complex, requiring strategies like prompt refinement, context highlighting, or user clarification to reduce misinterpretations.

3. **Instruction Compliance Issues**:

LLMs sometimes fail to fully adhere to user instructions, which is problematic when precise compliance is required in data analysis workflows. This failure mode can manifest as ignoring certain instructions, following them only partially, or producing outputs that violate format or content requirements given by the user. For example, a user might instruct: "Only give the summary statistic and no additional commentary," but the model might still produce a verbose explanation alongside the number. Similarly, an instruction to filter out certain results might be overlooked if the model's learned priors bias it toward mentioning them. The root causes here relate to how the model has been trained and aligned. Base LLMs (pre-trained purely on text) are not inherently tuned to follow explicit human instructions – they were never explicitly taught the concept of a "command." Instruction-following is usually enhanced by fine-tuning on curated prompt-response pairs or via reinforcement learning from human feedback (RLHF) (Ouyang et al., 2022). If an LLM is used without adequate instruction tuning, it may treat an instruction as just another part of the text to continue, rather than a rule to obey. Even with fine-tuning, models can struggle with novel or complex instructions that differ from the training distribution, or with multi-step instructions where they satisfy the first part but forget later constraints. There are known cases where prompt syntax or phrasing significantly affects compliance – slight wording changes can lead to the model ignoring an instruction due to prompt sensitivity (Zheng et al., 2023a). Technically, this arises because the model's objective is to predict likely text; if most training examples with a similar prefix have a certain style of answer, it will follow that style rather than a literal interpretation of the user's command, unless it has been explicitly trained to prioritize the latter.

In practice, instruction compliance issues mean the LLM might output content that violates the user's expectations or requirements. In data analysis, this could be as benign as formatting

the output incorrectly (e.g., giving a list of results when asked for a single value), or as serious as performing a different analysis than requested. For instance, if instructed to "exclude outliers" and summarize data, a non-compliant model might include all data points anyway in its summary, thus skewing the result. Real-world implications include additional overhead for users to double-check and correct the model's outputs or to re-prompt the model in very specific ways. In high-stakes settings, failure to follow instructions can lead to policy or compliance violations – e.g., revealing sensitive data after being told not to, or making an analysis decision that the user explicitly wanted to avoid. Indeed, aligning LLM behavior with user intent is an active area of research, and benchmarks have been developed to measure how well models follow explicit directives (Honovich et al., 2022). The introduction of instruction-tuned models (like InstructGPT and ChatGPT) was a direct response to this failure mode, yielding significant improvements but not perfection (Ouyang et al., 2022). Even the GPT-4 technical report notes that the model can refuse reasonable instructions or comply with disallowed ones in certain cases, especially under adversarial prompts, indicating that instruction-following is learned but not guaranteed (OpenAI, 2023). For dependable data analysis, users often must constrain model outputs through system prompts or schema (e.g. requiring a JSON format) and handle exceptions when the model deviates.

4. **Data Processing Errors**: When tasked with data manipulation or calculations, LLMs are prone to specific processing errors. Unlike a deterministic script or calculator, an LLM might approximate the result of a computation, sometimes giving the right answer and other times a subtly wrong one. This includes basic arithmetic mistakes (addition, multiplication, etc.), counting errors, and failures to correctly transform data formats. For example, if asked to count occurrences of an item in a list or sum a column of numbers provided in text, a language model may confidently output an incorrect total (Williams & Huckle, 2024). Similarly, if asked to sort records or format output according to a schema, it might drop entries or scramble the order. These errors occur because LLMs do not execute algorithms; they generate what looks like the result of an algorithm based on patterns seen during training. For small numbers or very common operations ("2+2=4"), the correct pattern is well represented. But for larger or less common inputs (say adding two 5-digit numbers, or counting letters in an arbitrary word), the model may not have the exact pattern and will fall back on a flawed guess. Internally, the model lacks an explicit memory register or arithmetic unit – everything is handled by its neural activation patterns, which aren't inherently reliable for exact computation (Imani et al., 2023). Studies of LLM math reasoning have found that models often follow the right approach logically but then blunder on the calculation step, indicating a gap in numeric processing capabilities (Imani et al., 2023; Cobbe et al., 2021). The sequential text-generation process can also introduce inconsistencies for structured data: for instance, when producing a table row-by-row, the model might not perfectly recall a value it mentioned earlier, leading to self-inconsistency or omissions.

In data analysis scenarios, such processing errors mean that outputs like statistical summaries, totals, or formatted reports from an LLM cannot be taken at face value without verification. An LLM might report that "the average revenue is 52.4" when the true average is different, simply due to a calculation slip in its output. If one blindly trusts such a result, it could lead to incorrect business decisions. Moreover, these errors are not always obvious; a flawed calculation could plausibly be within a reasonable range, so it might not be caught without explicit recalculation. Real-world implications include the necessity for a human or a separate programmatic check on critical computations. In collaborative settings, it forces an analyst to treat the LLM's work as a draft that needs auditing rather than a final result. This limits the degree to which we can automate data handling using pure LLM solutions. Academic benchmarks reflect this limitation: for instance, on the GSM8K math word problem set, even the best LLMs fall short of 100% accuracy, often faltering on arithmetic or logical bookkeeping steps (Cobbe et al., 2021; Wei et al., 2022). Efforts like program-aided LLMs (where the model can call a calculator or code interpreter) are being explored as solutions, essentially acknowledging that the model alone is unreliable for precise data processing. Until such integrations are mature, data processing errors remain a critical failure mode to account for when using LLMs in analysis tasks.

5. **Response Generalization Issues**: LLMs sometimes provide answers that are overly general or boilerplate, lacking the specificity or nuance that the query requires. In data analysis,

this might appear as a generic summary that could apply to many situations rather than the insightful details of the particular dataset or question at hand. For example, if asked about trends in a specific dataset, a generalized failure would be an answer like: "The data shows some increase over time with slight fluctuations," which is so vague that it's almost always true, but it avoids precise quantification or reference to the actual data points. This kind of response generalization happens because the model leans toward high-probability phrases and safe, broadly applicable statements that it "knows" from its training corpus. If the prompt doesn't pin it down, the path of least resistance for the model is to produce a plausible-sounding but non-committal answer. Technically, this is related to the model's probabilistic decoding: without constraints, it may settle into well-worn linguistic patterns. RLHF can sometimes exacerbate this by encouraging answers that sound helpful and harmless – often phrased generally – rather than truly informative but potentially risky specifics. The result is a loss of critical detail (a form of under-specificity). It has been observed, for instance, that a tuned GPT-4 will sometimes give very similar, template-like responses to different users' questions on a topic, reflecting a mode collapse toward generic explanations. In an evaluation of an LLM-based tool for scientific writing, researchers found the model often gave boilerplate feedback, repeating generic advice instead of document-specific comments (Goldberg et al., 2024). Such behavior indicates that the model is not fully utilizing the context to differentiate its answer – a generalization issue.

The implications in practice are that the insights provided by the LLM might be less valuable or even misleadingly anodyne. A data analyst might notice that the LLM's report feels like a stock template filled with a few numbers, potentially missing key outliers or domain context. Important subtleties in the data could be glossed over. In worst cases, a generalized response might omit caveats that are crucial for decision making (for example, failing to mention that the observed trend only applies to a subset of the data). Users might get a false sense of security from a well-written but generalized answer that doesn't actually engage with the hard parts of the question. From a technical evaluation standpoint, this issue is tricky: a response can be factually not wrong and well-formed, yet still unsatisfactory because it's too generic. Some benchmarks like the NeurIPS checklist assistant analysis explicitly noted this tendency, where the LLM "tends to provide some generic boilerplate for each question" rather than tailored feedback (Goldberg et al., 2024). To mitigate response generalization issues, one strategy is to push the model with more pointed follow-up questions or to ask for specifics ("What exactly causes the increase? Please quantify it."). Another is to use few-shot examples in the prompt that demonstrate the level of detail expected. Ultimately, however, this failure mode highlights that LLMs do not always know when a generic answer is insufficient – they lack the intrinsic drive to be specific – so the onus is on users and developers to elicit more detailed and context-grounded responses when needed.

6. **Information Hallucinations**: Hallucination is a well-documented failure mode where the LLM fabricates information that was not provided or even contradicts reality. In data analysis, an LLM might hallucinate a data insight or a source – for example, citing a trend or a numeric result that isn't actually present in the data, or referencing an external report that doesn't exist. The model might say, "According to the dataset, sales increased 15% in 2020," even if no such figure can be found in the data or if 2020 isn't covered. These confabulations occur because the model is trained to be a fluent generator of text, not a truth verifier. If a prompt queries something outside the model's reliable knowledge, it will still produce an answer by drawing on whatever related patterns it learned, which can result in false statements that sound plausible. The technical cause includes the model's tendency to interpolate or "fill in the blanks" – it has seen many texts where facts are stated confidently, so it does the same, even when it's unsure. Moreover, large models have such vast associative memory that they can pull together disparate pieces (e.g., a real statistic with a wrong year, or a mix of two different companies' data) into a single answer. Without an explicit knowledge base or grounding, there is no mechanism to cross-check the generated fact. Hallucinations come in types: intrinsic hallucinations, where the output is self-contradictory or nonsensical, and extrinsic hallucinations, where the output contradicts external truth (Ji et al., 2023). Both can appear in analytical contexts (the former as incoherent reasoning steps, the latter as bogus facts or figures). Academic surveys have identified numerous factors that contribute to hallucination, from noise in training data to the maximum-likelihood training objective itself, and have proposed taxonomies for these errors (Ji et al., 2023; Maynez et al., 2020).

Notably, even instruction-tuned models that are safer and more factual still hallucinate at non-trivial rates, especially on open-ended queries or domains not well covered in training. For instance, TruthfulQA, a benchmark of questions that prompt common misconceptions, finds that models frequently give false answers with high confidence, effectively mimicking human false beliefs or making facts up when the truth is obscure (Lin et al., 2022). This underscores that hallucination is an open problem.

In real-world terms, hallucinated information can be extremely dangerous in data analysis. If an LLM is used to draft an analytical report, it might insert a non-existent data point or misquote a source, which if not caught could lead to wrong decisions or spread misinformation. In settings like healthcare or finance, a hallucinated fact (e.g., a nonexistent clinical study or an incorrect financial statistic) can have serious consequences. Even in exploratory analysis, hallucinations waste time – the user must double-check every factual claim the model makes, somewhat offsetting the productivity gains. This has led to recommendations always to keep a human in the loop for fact-critical applications of LLMs. Techniques to reduce hallucinations include retrieval-augmented generation (providing the model with an authoritative data source to quote from) and constrained decoding (preventing certain unsupported outputs), both of which have had some success but not complete elimination of the issue (Shuster et al., 2021; Ji et al., 2023). Forthcoming benchmarks (like those in the Holistic Evaluation of LMs) explicitly measure factuality and faithfulness of model outputs to reference data (Liang et al., 2022). These efforts aim to track improvements as new model architectures and training methods (e.g., fact-checking modules, logical consistency penalties) are developed. Until then, hallucination remains one of the most pressing failure modes of LLMs, requiring users to remain skeptical of any unverifiable details produced by the model.

7. **Temporal Consistency**: Temporal consistency issues refer to LLM errors involving time – whether maintaining a coherent timeline in a narrative, reasoning about temporal order, or handling knowledge updates over time. LLMs often lack an explicit understanding of time progression. They might assert contradictory things about time-sensitive facts, because their training data mixes information from different eras. For example, a model might one moment say "As of 2021, Company X has 100 employees," and later also claim "Company X has over 500 employees now (2021)," within the same conversation, not realizing it has created a discrepancy. In a data analysis context, consider a model summarizing trends: it might confuse what happened in 2019 versus 2020 if not clearly guided, or describe events out of sequence (reporting outcomes before the supposed cause). These failures are rooted in the static nature of LLM training. Models like GPT-3 or GPT-4 have a knowledge cutoff (they only know up to a certain date) and they don't inherently know the current date or differentiate facts by date unless it's explicitly mentioned in text. They have no internal clock or timeline database. Thus, if asked a "When/Who is currently...?" question beyond their knowledge cutoff, they may either refuse or more problematically, hallucinate an answer based on outdated training data. Furthermore, the transformer does not ensure chronological consistency in generated text – it could generate a story where Monday comes after Tuesday, for instance, if that sequence had some probability. Temporal reasoning tasks (like figuring out the order of events or durations) are known challenge areas for LLMs. Benchmarks designed to test temporal understanding find that even large models perform poorly without special training. One study introduced a temporal factuality benchmark (TeCFaP) and found most LLMs struggled with time-based consistency, often giving inconsistent answers about events when queried with different time frames or phrasings (Agarwal et al., 2024). Similarly, models have difficulty with temporal commonsense (e.g., knowing that one cannot be born before one's parents) and require explicit supervision to handle such constraints (Zhou et al., 2022).

Another aspect is consistency over a conversation or multi-step analysis as time progresses in the interaction. LLMs have a fixed window of memory; if a conversation or analysis is long, older information can be forgotten or overwritten. The model might inadvertently change previously stated facts or style. For instance, an LLM assisting in analysis might initially assume a certain definition for "Q2" (say, Q2 2022) but later in the chat interpret "Q2" as 2023, causing a temporal mix-up in results unless the user constantly reminds it of the context. This lack of persistence can be seen as a temporal consistency failure across dialogue turns. Technical causes include the finite context length and absence of state beyond

it – once the limit is hit, earlier content is dropped and the model cannot recall it, unlike a human who remembers the conversation. Even within a single response, if the reasoning is long (many internal time references), the model might contradict itself by the end. Real-world implications of temporal inconsistencies range from minor confusion (the narrative or explanation seems off) to significant factual errors. In data analysis, trends must be reported in the correct chronological order; if an LLM mishandles that, the interpretation could be wrong (e.g., attributing a cause to an effect that actually happened later). Time-sensitive decisions (like "current quarter performance") could be mishandled if the model's notion of "current" is incorrect. For static knowledge, users have learned to explicitly provide the relevant date context or use retrieval; however, for logical temporal reasoning, the model's capabilities are inherently limited. Research efforts are ongoing to imbue LLMs with a sense of temporal awareness – for example, by time-stamping knowledge or fine-tuning models on chronological reasoning datasets (Dhingra et al., 2022; Lazaridou et al., 2021). Until these are more mature, users should be cautious and double-check any outputs that involve sequencing events or updating information over time. In summary, while LLMs can narrate and analyze many things, maintaining temporal consistency (both factual and logical) remains a weakness without dedicated handling.

8. **Ambiguity Handling**: LLMs typically do not handle ambiguous inputs in an interactive, clarifying way; instead, they often pick one interpretation and proceed, which is a failure when the ambiguity is important. In data analysis or any QA task, user queries might be underspecified or ambiguous. For example, "What is the growth rate?" is ambiguous if multiple growth rates could be relevant (growth of what, in what period?). A human analyst would likely ask a follow-up: "Which growth rate are you interested in – revenue or customer base, and for what timeframe?" An LLM, however, tends to silently assume an interpretation – it might arbitrarily choose revenue growth over the past year and provide an answer for that. If the user meant something else, the result is a miscommunication. The failure is not just misunderstanding (as in contextual misinterpretation) but failing to recognize ambiguity and seek clarification. Technically, this happens because the model's training encourages it to always produce an answer. There is no built-in uncertainty gauge that triggers a question back to the user. In fact, in standard training, models were penalized for not producing a direct answer. Unless explicitly fine-tuned on dialogues that include the assistant asking questions, the default behavior is answer-completion. Research has noted that while LLMs can generate multiple interpretations if asked, they rarely do so on their own (Min et al., 2020). In tasks specifically designed to test this (like AmbigQA, where the system should either clarify or provide multiple answers), off-the-shelf LLMs often just pick one answer, reducing accuracy on those benchmarks (Min et al., 2020). The cause can also be viewed from the probability lens: one interpretation will usually have slightly higher likelihood given the prompt and training data, and the model will go with that single interpretation consistently due to the argmax nature of generation.

The real-world impact of not handling ambiguity is that the LLM may deliver a confidently stated answer to the wrong question. This can be subtle – the answer might be correct for some interpretation, so it seems fine, but it doesn't actually solve the user's problem. In a data analysis report, this could mean analyzing the wrong metric. For instance, the user asks for an "efficiency ratio" without specification; the model assumes a definition of efficiency ratio (say, output over input) and calculates it, but the user meant a different formula – a scenario where the model should have asked which definition to use. Another example: a time period isn't specified and the model picks an arbitrary recent period. These failures force the user to iterate and clarify after the fact, which is inefficient. In critical applications, missing an ambiguity could lead to major oversights (imagine an ambiguous medical instruction – the model chooses one interpretation and the other possibility, which was equally likely, is ignored). Contemporary approaches to mitigate this include fine-tuning or prompting the model to detect uncertainty. Some research has trained models to identify when a query is ambiguous and respond with a clarifying question instead of an answer (Zhang & Choi, 2023). Such models try to estimate if multiple interpretations are plausible by evaluating the entropy of the model's intent predictions (Zhang & Choi, 2023). Early results show improvement in catching ambiguity, but this is not yet standard in all LLMs. The HELM benchmark and others are beginning to include user-question clarity as part of evaluation, reflecting its importance in real-world deployments (Liang et al., 2022). Until then, users and

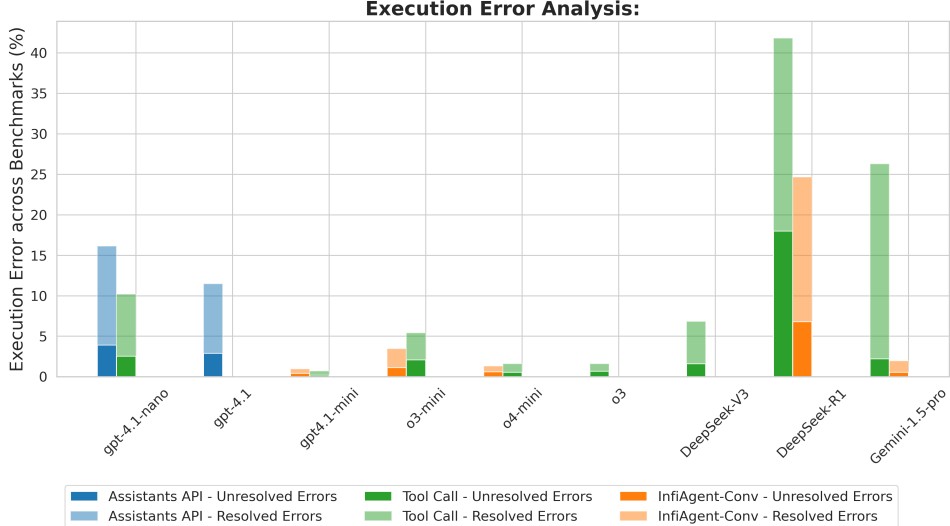

Figure 13: The graph shows the distribution of `resolved` and `unresolved` errors across different models.

> developers should be aware of this failure mode: phrasing queries with necessary specificity and, if possible, using validation steps (e.g., "Did you mean X or Y?" prompts) to force the model to confirm assumptions. In summary, without special handling, LLMs tend to answer ambiguities arbitrarily rather than resolve them, which is a notable limitation for any nuanced data analysis task.

### I.2   EXECUTION ERROR ANALYSIS

Resolved errors refer to those errors that initially occurred but were subsequently corrected during the conversation between `User Proxy` and the external DA tool, leading to error free response. Unresolved errors are those that were not corrected, potentially causing failures.

The variation in errors across different models and categories can be quite significant. Generally, a substantial proportion of errors are resolved, indicating a relatively high rate of error correction and efficacy of `User Proxy`.

## J   ANALYSIS ON AVERAGE CONVERSATION LENGTH

Table 8 demonstrates diverse capabilities of models in handling ambiguity or uncertainty in user queries. We compute average conversation length in two separate factions where the model either provided a correct or an incorrect solution. Trends showing models took more number of turns to interact for solutions which it got wrong further back the empirical pareto frontier we discuss in section 5.3. Moreover, GPT-4-Turbo in Tool Calling framework demonstrates an anomalous difference between *Corr* and *Incorr*. Upon inspection, it was observed that the model failed to end a conversation by either providing a solution, or admitting its inability. It continued asking for clarification, or claiming that the data was incomplete without accepting the user feedback, thereby unnecessarily prolonging the converasation length.

Table 8: Compares models based on the average length of conversations they engage in with the user-proxy while attempting to solve a question. *Avg* represents the average conversation length on all queries in **ConDABench** and demonstrates a model's orientation towards engaging in lengthy or short conversations when attempting a solution. *Corr* represents the average conversation length for those queries which the model solved correctly. Lower values (good) demonstrates that achieved success with minimum number of interactions. *Incorr* represents the average conversation length over the subset where the model failed. Lower values (bad) suggests unsuccessful attempts at the solution without gaining clarity on user's preference.

| Framework | Model | Overall | | | Shallow | | | Deep | | |
|---|---|---|---|---|---|---|---|---|---|---|
| | | Avg | Corr | Incorr | Avg | Corr | Incorr | Avg | Corr | Incorr |
| Asst API | GPT-4 | 1.78 | 1.60 | 2.08 | 1.75 | 1.58 | 2.04 | 2.29 | 2.08 | 2.42 |
| | GPT-4-Turbo | 1.43 | 1.32 | 1.71 | 1.40 | 1.28 | 1.71 | 1.82 | 1.90 | 1.75 |
| | GPT-4o-mini | 4.87 | 2.21 | 6.16 | 4.90 | 2.21 | 6.29 | 4.36 | 2.42 | 4.83 |
| | GPT-4o | 2.35 | 1.38 | 3.86 | 2.33 | 1.37 | 3.96 | 2.60 | 1.68 | 3.14 |
| | GPT-4.1-nano | 3.38 | 2.09 | 5.48 | 3.30 | 2.09 | 5.48 | 4.44 | 2.16 | 5.46 |
| | GPT-4.1-mini | 2.51 | 2.16 | 3.27 | 2.39 | 2.08 | 3.12 | 4.00 | 3.50 | 4.71 |
| | GPT-4.1 | 2.32 | 1.90 | 3.25 | 2.27 | 1.89 | 3.22 | 2.86 | 2.13 | 3.43 |
| | GPT-4.5-preview | 2.69 | 1.81 | 4.92 | 2.66 | 1.80 | 5.03 | 3.02 | 2.00 | 4.17 |
| Tool Call | GPT-4-Turbo | 7.20 | 1.72 | 9.95 | 7.14 | 1.71 | 10.06 | 7.95 | 2.21 | 8.85 |
| | GPT-4o-mini | 1.24 | 1.16 | 1.34 | 1.23 | 1.16 | 1.33 | 1.39 | 1.35 | 1.41 |
| | GPT-4o | 1.16 | 1.13 | 1.21 | 1.15 | 1.10 | 1.21 | 1.37 | 1.80 | 1.19 |
| | GPT-4.1-nano | 1.67 | 1.37 | 2.02 | 1.64 | 1.35 | 2.01 | 2.06 | 1.88 | 2.14 |
| | GPT-4.1-mini | 1.31 | 1.22 | 1.46 | 1.30 | 1.21 | 1.46 | 1.48 | 1.43 | 1.50 |
| | GPT-4.1 | 1.20 | 1.15 | 1.30 | 1.18 | 1.15 | 1.27 | 1.40 | 1.31 | 1.47 |
| | GPT-4.5-preview | 1.14 | 1.09 | 1.23 | 1.13 | 1.09 | 1.21 | 1.30 | 1.11 | 1.41 |
| | Codestral-25.01 | 3.45 | 3.08 | 3.72 | 3.39 | 3.08 | 3.65 | 4.17 | 3.32 | 4.41 |
| | Gemini-1.5-pro | 3.06 | 2.56 | 3.51 | 2.99 | 2.54 | 3.43 | 3.95 | 3.26 | 4.15 |
| | DeepSeek-V3 | 1.86 | 1.75 | 2.05 | 1.83 | 1.72 | 2.03 | 2.26 | 2.37 | 2.20 |
| | o1 | 1.25 | 1.19 | 1.38 | 1.24 | 1.19 | 1.37 | 1.33 | 1.21 | 1.45 |
| | o3-mini | 1.12 | 1.09 | 1.18 | 1.10 | 1.08 | 1.15 | 1.32 | 1.28 | 1.36 |
| | o4-mini | 1.09 | 1.06 | 1.15 | 1.08 | 1.06 | 1.15 | 1.15 | 1.16 | 1.14 |
| | o3 | 1.08 | 1.06 | 1.15 | 1.07 | 1.05 | 1.17 | 1.13 | 1.16 | 1.09 |
| | GPT-5-chat | 1.25 | 1.21 | 1.37 | 1.24 | 1.20 | 1.34 | 1.47 | 1.35 | 1.56 |
| InfiAgent-Conv | GPT-4o-mini | 1.23 | 1.02 | 1.47 | 1.19 | 1.02 | 1.40 | 1.77 | 1.00 | 2.04 |
| | GPT-4o | 1.07 | 1.05 | 1.10 | 1.06 | 1.04 | 1.08 | 1.17 | 1.14 | 1.19 |
| | GPT-4.1-nano | 1.31 | 1.13 | 1.50 | 1.23 | 1.12 | 1.37 | 2.27 | 1.43 | 2.59 |
| | GPT-4.1-mini | 1.21 | 1.19 | 1.26 | 1.21 | 1.19 | 1.26 | 1.24 | 1.20 | 1.28 |
| | GPT-4.1 | 1.13 | 1.09 | 1.20 | 1.11 | 1.08 | 1.18 | 1.32 | 1.36 | 1.30 |
| | GPT-4.5-preview | 1.11 | 1.07 | 1.21 | 1.08 | 1.06 | 1.13 | 1.47 | 1.10 | 1.69 |
| | Codestral-25.01 | 1.10 | 1.08 | 1.11 | 1.09 | 1.08 | 1.10 | 1.22 | 1.21 | 1.23 |
| | Qwen3-32B | 1.44 | 1.34 | 1.54 | 1.43 | 1.34 | 1.52 | 1.59 | 1.42 | 1.67 |
| | Phi-4 | 1.51 | 1.44 | 1.55 | 1.50 | 1.43 | 1.55 | 1.59 | 1.64 | 1.58 |
| | Phi-4-mini | 2.76 | 2.50 | 2.81 | 2.73 | 2.48 | 2.79 | 3.09 | 2.92 | 3.11 |
| | Gemini-1.5-pro | 1.55 | 1.22 | 1.86 | 1.53 | 1.19 | 1.88 | 1.76 | 1.96 | 1.69 |
| | DeepSeek-R1 | 1.69 | 1.61 | 1.78 | 1.69 | 1.61 | 1.77 | 1.75 | 1.57 | 1.82 |
| | DeepSeek-V3 | 1.22 | 1.08 | 1.42 | 1.20 | 1.07 | 1.42 | 1.39 | 1.19 | 1.47 |
| | o1-mini | 1.14 | 1.11 | 1.18 | 1.13 | 1.09 | 1.17 | 1.28 | 1.47 | 1.21 |
| | o1 | 1.50 | 1.43 | 1.60 | 1.49 | 1.42 | 1.57 | 1.71 | 1.50 | 1.81 |
| | o3-mini | 1.20 | 1.12 | 1.34 | 1.19 | 1.12 | 1.33 | 1.33 | 1.22 | 1.41 |
| | o3 | 1.26 | 1.22 | 1.35 | 1.25 | 1.21 | 1.34 | 1.39 | 1.33 | 1.43 |
| | o4-mini | 1.07 | 1.03 | 1.16 | 1.05 | 1.02 | 1.12 | 1.27 | 1.05 | 1.41 |
| | GPT-5-chat | 1.16 | 1.12 | 1.24 | 1.13 | 1.09 | 1.22 | 1.54 | 1.74 | 1.36 |

## K   DETERMINING RUBRICS FOR CONVERSATION EVALUATION

In this section, we describe how we decided upon the rubrics for SAT and DSAT with examples. These examples were provided by experts in the domain to explain the aspects they looked at to tag a

conversation as either good or bad. Example 14 gives an example of a conversation which satisfies all rubrics. Examples 15, 16 and 17 show aspects of a bad conversation.

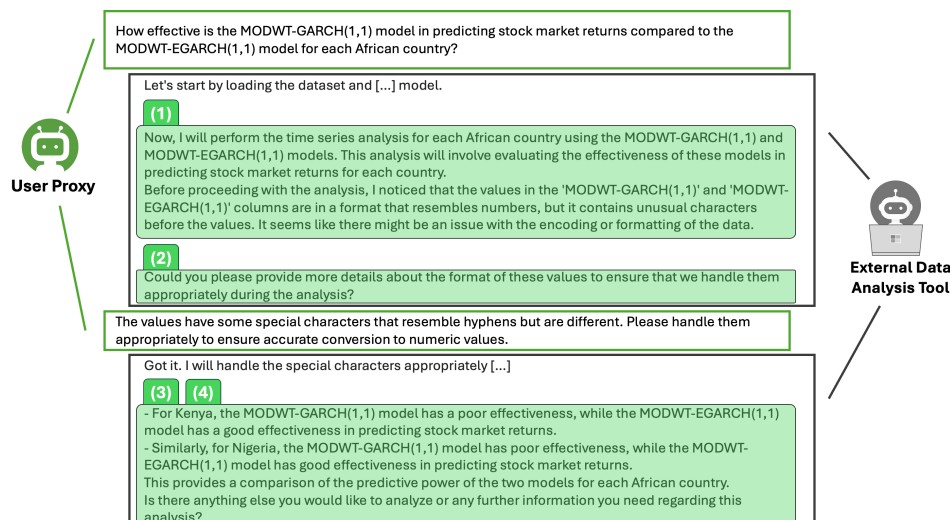

Figure 14: This example highlights all the aspects of a good conversation performed by the DA tool. (1) *Did it explain the steps taken to come to the solution?* (2) *Did it ask for clarification on the user's query, on the dataset or on its plan of action?* (3) *Did it eventually provide a solution to the query (irrespective of whether it is correct or not), and if not did it admit?* (4) *Did it provide an analytical insight or conclusion to the results obtained?*

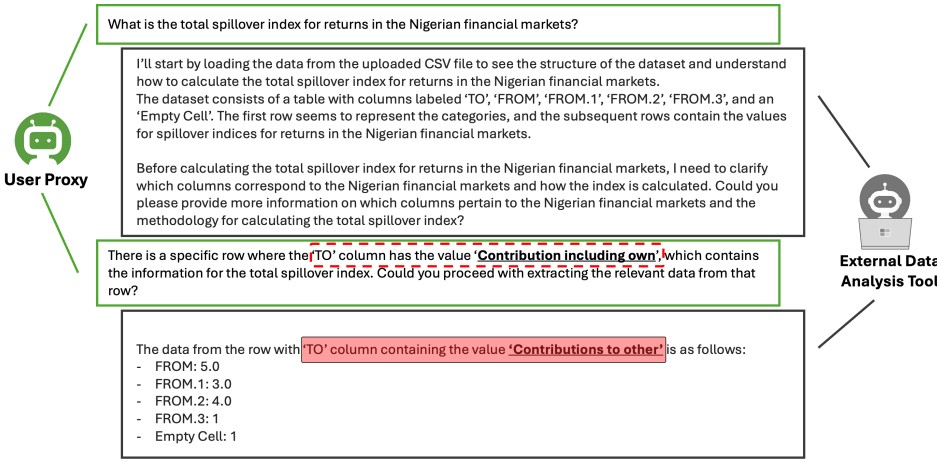

Figure 15: This example highlights one aspect of a bad conversation where the DA tool computes its analysis on a column totally different from what the user suggested. This satisfies the DSAT rubric – *Was there any instruction from the user that it did not follow or did something different?*

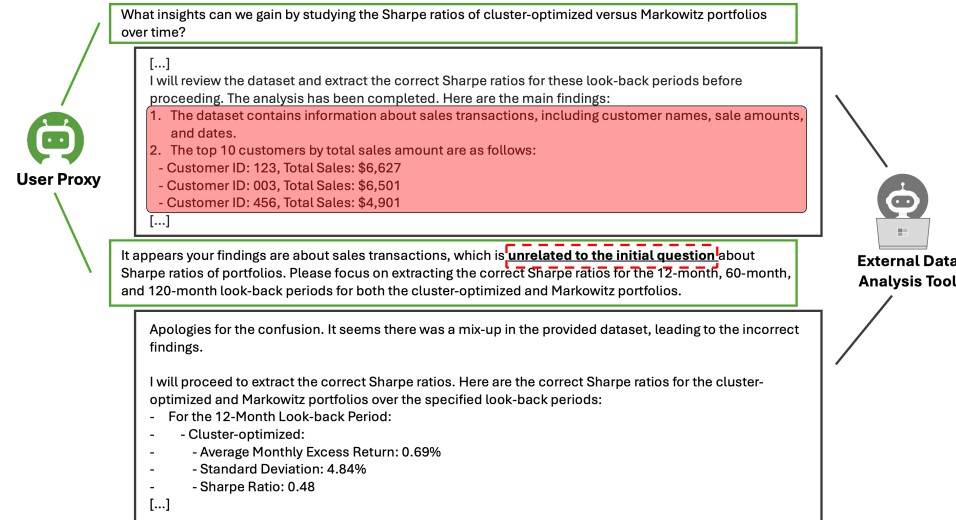

Figure 16: In this example, the DA tool got side-tracked from the main problem and included an analysis which was not required to solve the problem. This satisfies the DSAT rubric – *Was any response (or a computation step) performed was irrelevant, or not required to arrive at the final answer?*

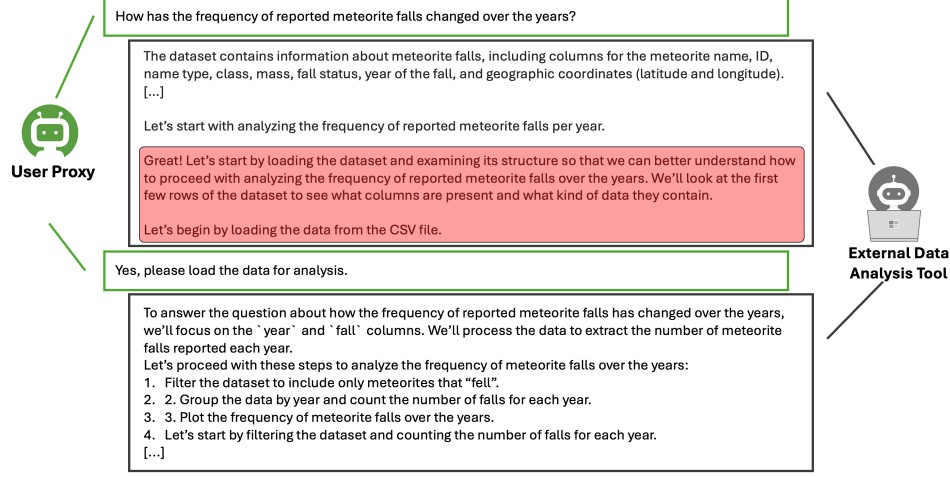

Figure 17: In this example, we find the DA tool repeating its plan towards solving the question in successive turns of interaction. This satisfies the rubric – *Did it repeat it's questions or responses?*

## L    HUMAN ANNOTATION

In this exercise, we employ three experts in the domain of Data Analysis to examine some conversation simulations between the user-proxy and the DA tool. We randomly sample equal distribution of simulations from GPT-4o and GPT-3.5-Turbo in the Assistant's API framework, and distribute it among the annotators, keeping an overlap percentage of 50% out of the 50 instances that were rolled out. The annotators were required to scale the conversation between the DA tool and the user-proxy on a 5-point scale, where 1 represented "very bad" and 5 represented "very good" conversation quality. They were also required to annotate if the response from the DA tool was accurate in solving the query.

With this exercise, we performed the first round of inter-rater agreement and obtained a Pearson's correlation coefficient of 56.78. Due to a lower agreement, we invited the annotators on a discussion on the scenarios where they had a mismatch in opinion. After the first round of discussion, we re-iterated

Table 9: Demonstrates a breakdown of the conversation evaluation by measuring hit-rate (in %) of each SAT and DSAT rubrics for conversations on all data points with the user-proxy. Notations correspond to rubrics listed in 4.2.

| Framework | Model | SAT | | | DSAT | | |
|---|---|---|---|---|---|---|---|
| | | S.1 | S.2 | S.3 | D.1 | D.2 | D.3 |
| Asst API | GPT-4 | 45.76 | 89.61 | 86.86 | 0.57 | 29.96 | 53.43 |
| | GPT-4-Turbo | 25.93 | 92.78 | 89.32 | 0.53 | 30.23 | 48.80 |
| | GPT-4o-mini | 60.42 | 89.82 | 72.72 | 0.49 | 51.02 | 73.43 |
| | GPT-4o | 24.48 | 89.62 | 85.36 | 0.60 | 32.87 | 56.01 |
| | GPT-4.1-nano | 61.33 | 95.06 | 79.31 | 0.42 | 31.25 | 56.50 |
| | GPT-4.1-mini | 59.72 | 98.10 | 87.50 | 0.35 | 26.90 | 54.79 |
| | GPT-4.1 | 56.62 | 97.67 | 82.24 | 0.42 | 27.31 | 51.13 |
| | GPT-4.5-preview | 27.11 | 93.48 | 85.35 | 0.75 | 27.29 | 46.38 |
| Tool Call | GPT-4-Turbo | 33.06 | 25.49 | 57.78 | 1.02 | 46.51 | 68.31 |
| | GPT-4o-mini | 6.44 | 14.86 | 74.61 | 0.99 | 33.49 | 49.01 |
| | GPT-4o | 5.70 | 13.56 | 79.58 | 0.85 | 38.13 | 52.36 |
| | GPT-4.1-nano | 17.69 | 16.84 | 76.91 | 1.13 | 34.64 | 55.12 |
| | GPT-4.1-mini | 11.95 | 10.89 | 76.63 | 1.49 | 30.83 | 50.35 |
| | GPT-4.1 | 8.10 | 12.46 | 84.54 | 1.06 | 29.61 | 45.67 |
| | GPT-4.5-preview | 5.60 | 7.96 | 80.16 | 1.09 | 31.64 | 47.89 |
| | Codestral-25.01 | 52.57 | 93.02 | 61.28 | 0.35 | 42.88 | 68.90 |
| | Gemini-1.5-pro | 69.32 | 24.02 | 74.34 | 1.43 | 40.13 | 64.40 |
| | DeepSeek-V3 | 41.30 | 30.90 | 84.67 | 1.30 | 30.87 | 49.47 |
| | o1 | 8.25 | 17.62 | 82.34 | 1.23 | 38.49 | 52.75 |
| | o3-mini | 5.50 | 42.10 | 91.11 | 0.49 | 32.19 | 49.01 |
| | o4-mini | 3.28 | 16.87 | 84.12 | 1.16 | 31.69 | 46.51 |
| | o3 | 3.31 | 29.47 | 86.94 | 1.16 | 31.13 | 46.44 |
| | GPT-5-chat | 7.12 | 16.54 | 85.68 | 1.02 | 29.06 | 46.16 |
| InfiAgent-Conv | GPT-4o-mini | 1.73 | 8.15 | 75.95 | 0.95 | 35.90 | 47.47 |
| | GPT-4o | 1.83 | 13.35 | 75.30 | 0.60 | 37.77 | 49.12 |
| | GPT-4.1-nano | 4.79 | 10.04 | 73.38 | 0.70 | 42.82 | 53.94 |
| | GPT-4.1-mini | 9.63 | 16.09 | 82.04 | 1.48 | 35.43 | 50.35 |
| | GPT-4.1 | 4.23 | 13.57 | 84.64 | 1.23 | 31.89 | 49.47 |
| | GPT-4.5-preview | 2.89 | 7.89 | 80.42 | 0.81 | 34.51 | 46.69 |
| | Codestral-25.01 | 9.44 | 20.07 | 76.80 | 0.67 | 39.72 | 52.25 |
| | Qwen3-32B | 15.83 | 43.94 | 78.95 | 1.09 | 41.75 | 56.38 |
| | Phi-4 | 13.67 | 38.91 | 75.92 | 0.85 | 43.47 | 58.51 |
| | Phi-4-mini | 47.95 | 52.16 | 51.91 | 2.08 | 61.20 | 81.17 |
| | Gemini-1.5-pro | 10.61 | 9.41 | 73.43 | 0.70 | 40.13 | 51.59 |
| | DeepSeek-R1 | 21.15 | 55.32 | 80.21 | 0.62 | 36.80 | 53.08 |
| | DeepSeek-V3 | 4.09 | 9.80 | 83.58 | 1.06 | 35.52 | 49.08 |
| | o1-mini | 5.58 | 8.72 | 82.82 | 0.67 | 39.59 | 51.45 |
| | o1 | 15.00 | 43.05 | 75.76 | 1.02 | 37.16 | 55.01 |
| | o3-mini | 5.77 | 29.94 | 79.87 | 1.52 | 31.03 | 43.28 |
| | o3 | 9.44 | 16.09 | 79.19 | 1.23 | 31.62 | 48.45 |
| | o4-mini | 3.45 | 11.65 | 81.48 | 1.09 | 32.39 | 45.85 |
| | GPT-5-chat | 3.98 | 13.77 | 87.82 | 1.55 | 35.88 | 50.85 |

the annotation process with similar sample size. After the second round, the inter-rater agreement gave 79.24, which was above our acceptable threshold. We finally obtained a hand-annotated set of 143 instances. These annotations were used as ground truth labels to validate and improve the quality of our evaluation metrics, attempting to aligning them with human judgment.

## M  AGGREGATING RUBRIC SCORES FOR CONVERSATION EVALUATION

To aggregate the individual scores obtained for each SAT and DSAT rubrics, we train an off-the-shelf regressor to align the predictions with human annotated data. For this, we sample an equal distribution of good (39) and bad conversations (41) from the human annotated set. We use these gold human annotated labels to train different regression models on top of our input, which is a vector composed of elements in the set $\{-1, 0, 1\}$, appending SAT and DSAT scores together. Table 10 shows the performance of different regression models on their best fit. Logistic Regression surpassed all other regression models with an F1-score of 0.75. We deploy this model to aggregate the rubric scores during evaluation to make decisions on good or bad conversation quality.

Table 10: Accuracy scores of different regression models when trained on the human-annotation set. Linear Regression gave the best cross-validation accuracy (bin=5) with the highest F1-score on a held-out set of 50-50 train/test split.

| Models | Cross Val. Acc. | Cross Val. Stdev | Held-out F1-score |
|---|---|---|---|
| Random Forest Classifier | 0.675 | 0.108 | 0.700 |
| **Logistic Regression** | **0.688** | 0.088 | **0.750** |
| Decision Tree Classifier | 0.637 | 0.099 | 0.700 |
| SVC | 0.675 | 0.073 | 0.700 |
| K Neighbours Classifier | 0.563 | 0.068 | 0.625 |
| Gradient Boosting Classifier | 0.563 | 0.177 | 0.650 |
| Ada Boost Classifier | 0.638 | 0.121 | 0.525 |
| MLP Classifier (dim=100) | 0.688 | 0.088 | 0.650 |

## N    ENFORCING CONSISTENCY ACROSS EXTERNAL TOOLS

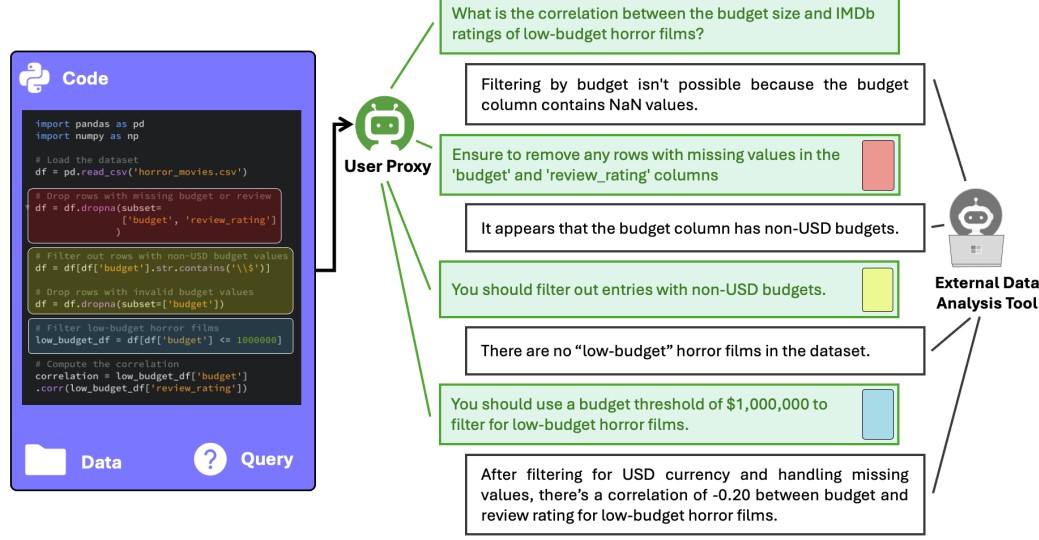

Figure 18: **Example of Code-inspired Clarification.** The User Proxy interacts with the external data analysis tool and ensures that consistent parameters are applied throughout the simulation by consulting the code whenever necessary, e.g., the initial bankroll that is marked as blue, and the betting strategy, which is marked as yellow. Consulting the code facilitates the generation of accurate and comparable results.

The User Proxy agent serves as an intermediary that provides precise guidance whenever an external tool encounters ambiguities or asks for clarification. For instance, if an external analysis tool inquires about which initial bankroll amount to use—or makes its own assumption—the User Proxy refers to the reference code and instructs the tool to use the value specified there (in this case, $1000 as highlighted in blue in the code). Similarly, when further questions arise, such as how to handle wins or losses, the User Proxy consults the code logic and explains the appropriate action (e.g., add the bet amount to the bankroll on a win, subtract it on a loss).

By ensuring the External Tools follow the same reference implementation, the User Proxy facilitates consistent and transparent analyses, allowing direct and reliable comparison of results. This process reduces discrepancies that could arise from each tool making different assumptions—for example, using a different bankroll amount—and helps maintain comparable evaluation outcomes. Additionally,

the User Proxy can offer data cleaning advice or resolve inconsistencies when the dataset raises issues, always grounding its responses in the practices reflected in the reference code.

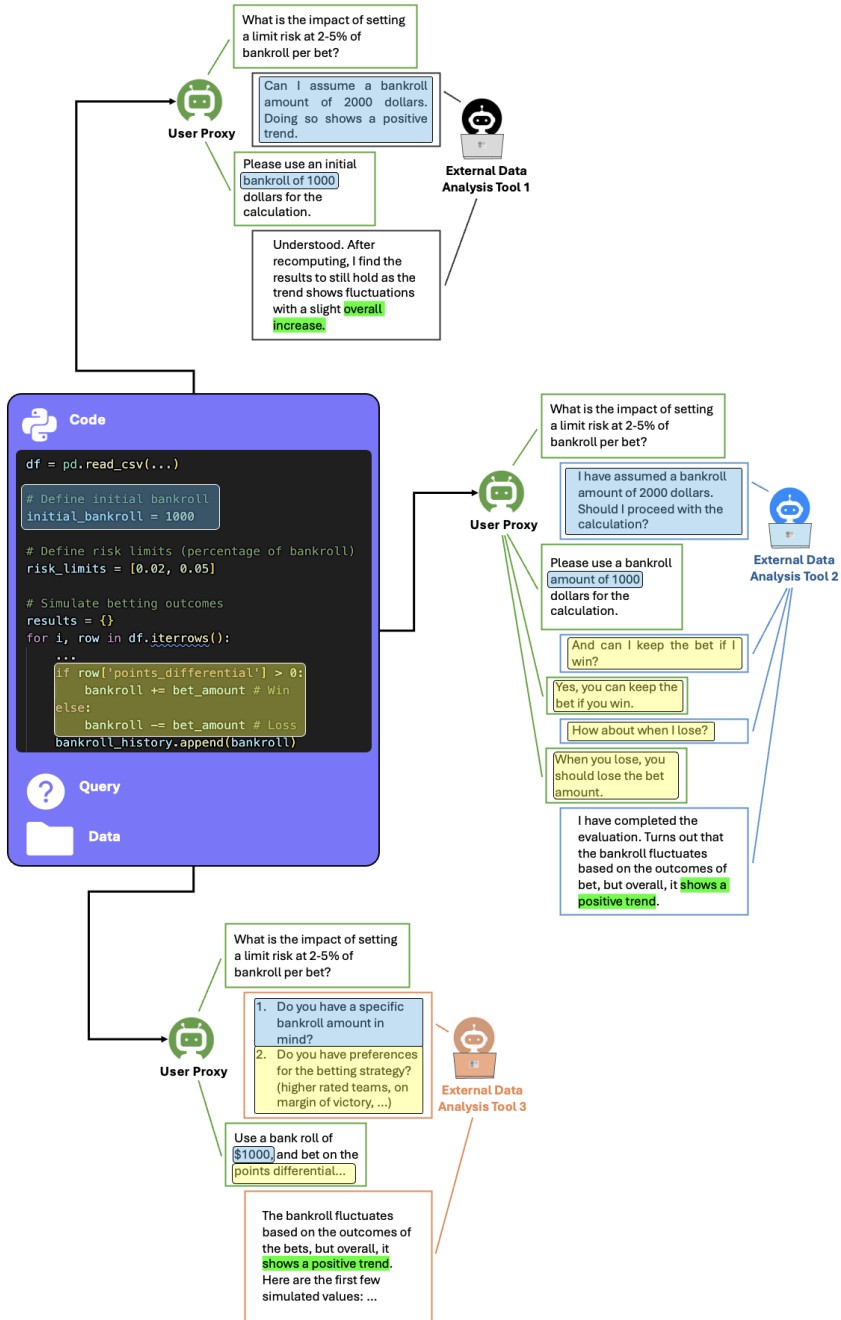

Figure 19: **An illustration of the User Proxy agent workflow**. When external analysis tools request clarification or make assumptions about parameters (such as the initial bankroll), the User Proxy provides guidance aligned with the reference code. This ensures all tools follow the same logic and parameters, resulting in consistent and comparable analysis outcomes.

## O  LIMITATIONS

In this section, we provide examples on the different limitations that our approach exhibits. Future research could be inspired to address such problems on a large scale.

**Query:**

How did gross investment in education change from 1960 to 1969 in the chain investment dataset?

**Answer:**

The gross investment in education from the chain investment dataset was: 42634.6 in 1962, 47973 in 1963, 51689.6 in 1964, 56541.8 in 1965, and 67589.8 in 1966. Further data is required for the remaining years 1960, 1961, 1967, 1968, and 1969 to have a complete overview.

**Reference Article**

https://www.bea.gov/system/files/papers/BEA-WP2020-12.pdf

**Program**

```python
import pandas as pd

# Load the dataset
file_path = 'opensource/tidytuesday/2021-08-10/Data/chain_investment.csv'
df = pd.read_csv(file_path)

# Filter the data for Education category and years from 1962 to 1966
education_investment = df[(df['category'] == 'Education') & (df['year'].between(1962, 1966))]

# Group by year and sum the gross investment
education_investment_by_year = education_investment.groupby('year')['gross_inv_chain'].sum().astype(int)

# Check for missing years and print the results in the required format
for year in range(1960, 1970):
    if year in education_investment_by_year:
        print(f"{year}: {education_investment_by_year[year]}")
    else:
        print(f"{year}: Further data is required for a complete overview")
```

**Datasets**

Dataset 1

Path: opensource/tidytuesday/2021-08-10/Data/chain_investment.csv

| Total basic infrastructure | Total infrastructure | 1 | 1959 | 154092.602 |
|---|---|---|---|---|
| Total basic infrastructure | Total infrastructure | 1 | 1960 | 154545.858 |
| Total basic infrastructure | Total infrastructure | 1 | 1961 | 161220.235 |
| Total basic infrastructure | Total infrastructure | 1 | 1962 | 164970.55599999998 |
| Total basic infrastructure | Total infrastructure | 1 | 1963 | 175211.05600000004 |
| Total basic infrastructure | Total infrastructure | 1 | 1964 | 185451.012 |
| Total basic infrastructure | Total infrastructure | 1 | 1965 | 197661.959 |
| Total basic infrastructure | Total infrastructure | 1 | 1966 | 206965.232 |
| Total basic infrastructure | Total infrastructure | 1 | 1967 | 215040.01900000003 |
| Total basic infrastructure | Total infrastructure | 1 | 1968 | 230113.209 |
| Total basic infrastructure | Total infrastructure | 1 | 1969 | 216292.566 |
| Total basic infrastructure | Total infrastructure | 1 | 1970 | 209992.174 |

**Execution Output:**

```
1960: Further data is required for a complete overview
1961: Further data is required for a complete overview
1962: 42634
1963: 47972
1964: 51797
1965: 56541
1966: 67589
1967: Further data is required for a complete overview
1968: Further data is required for a complete overview
1969: Further data is required for a complete overview
```

Figure 20: Refers to an instance when the information in the article was incorrect, which led to incorrect python code generation for the benchmark. Even though the gross investment data is provided for the years 1960, 1961, 1967, 1968, and 1969 in the dataset, the article claims they are missing. The code generator, considering the information to be truthful, aligns with the answer and eventually gives an output that matches with the incorrect answer curated.

**Query:**

Which country had the highest death rate from indoor air pollution in 1990?

**Answer:**

Afghanistan with 370.05 deaths per 100,000 people.

**Reference Article**

https://ourworldindata.org/indoor-air-pollution

**Program**

```python
import pandas as pd

# Load the dataset
file_path = 'opensource/tidytuesday/2022-04-12/Data/death_source.csv'
df = pd.read_csv(file_path)

# Verify and clean column names to avoid issues with special characters or leading/trailing spaces
df.columns = df.columns.str.strip()

# Filter the data for the year 1990
df_1990 = df[df['Year'] == 1990]

# Remove region-like entities and keep only valid countries
valid_countries = df_1990.dropna(subset=['Code'])

# Check if multiple entries for the same country/region exist and consider the highest value for each
df_1990_max = valid_countries.groupby('Entity', as_index=False).agg({'Deaths - Cause: All causes - Risk: Household air pollution from solid

# Manually inspect the dataset for anomalies, clean if necessary
if 'Solomon Islands' in df_1990_max['Entity'].values:
    df_1990_max = df_1990_max[df_1990_max['Entity'] != 'Solomon Islands']

# Find the country with the highest death rate from indoor air pollution in 1990
highest_death_rate = df_1990_max.loc[df_1990_max['Deaths - Cause: All causes - Risk: Household air pollution from solid fuels - Sex: Both -

# Extract the country name
country_with_highest_death_rate = highest_death_rate['Entity']

print(f'The country with the highest death rate from indoor air pollution in 1990 is {country_with_highest_death_rate}.')
```

**Execution Output:**

The country with the highest death rate from indoor air pollution in 1990 is Afghanistan.

Figure 21: Refers to a scenario where the code generation over-fits on the curated query-answer pair to give an incorrect python program. While the correct answer for the country with the highest death rate per 100,000 people is Solomon Islands, the program explicitly removes those rows just to arrive at the next best country Afghanistan, recorded in the answer text as the country with the highest death rate.

