# OpenReview forum: "ConDABench: Interactive Evaluation of Language Models for Data Analysis"
_ICLR.cc/2026/Conference — ICLR 2026 Conference Withdrawn Submission_

### Official Review · Reviewer_Dmpf · 2025-10-28

**Soundness:** 2
**Presentation:** 1
**Contribution:** 1
**Rating:** 2
**Confidence:** 4

**Summary:**

This paper introduces ConDABench, with a framework for generating and evaluating conversational data analysis benchmarks. The work tries to address a critical gap in existing benchmarks by providing first-class support for interactivity in data analysis tasks. The framework consists of three main components: (1) a multi-agent workflow for benchmark generation from real-world articles, (2) 1,420 ConDA problems across diverse query types, and (3) an evaluation harness with a code-grounded User Proxy agent for automated interactive evaluation.

**Strengths:**

1. Authors focus on a valuable question about iterative nature of real-world data analysis where queries are often vague and require clarification.
2. Authors provide additional and further metrics beyond benchmark like Tapilot-Crossing.
3. They also include APIs into account.

**Weaknesses:**

W1. This paper is quite incremental compared to existing work [1].

- The main motivation especially in paragraph 1 already exists in Introduction of [1], all use underspecific or ambiguous to illustrate essentials of interaction.
- Both of these papers include multi-agent workflow to generate data for benchmark, but [1] has a detailed section to evaluate it with human experts and a clear instructions  / principles of rating and evaluation. However, this paper only have one sentence in line 259 which makes the rigorousness of this benchmark questionable. Also [2-3] all have detailed explanation and judgement about bias evaluation for ai-based generated benchmark. Compared to these relevant work, this is an obvious weakness. The quality of benchmark is questionable.
- Code-grounded method is not new, in [1], they also have code generation and QA track, and their authors also produce QA based on executed results of codes. The workflow is also code-grounded. From my part, I don't see any other contributions compared to [1].Running example in 170 is even quite similar to statement in [1] (end of page 3 right column).
- Authors propose user-proxy agent, but [1] already contains that, is there any different?

W2. The writing is not good and hard to fully follow it:

 - Abstract mentions exisiting benchmarks do not capture "such complexities", which refer to "vauge and underspecific" previously mentioned. However, as far as I know, many benchmark include this features such as [1][4]. It's a very unclear statement.
- Then authors mention three contributions directly, but it seems there are no relations of how such contributions can solve "vague and underspecific" goals. The logics of abstract even make me quite confused.
- From the introduction, I do not see any new problems compared to existing works [1][2]. [1] is also a conversational data analysis benchmark which was made also by multi-agent workflow, but verified by rigorous and comprehensive human expert evaluation. They are also code-grounded and have user agent for under-specific part.
- I don't know what is the point of Section 2, it seems authors repeat the motivation, if so, it seems lengthy, they should put them Into troduction.

W3. The whole system is based on AI:

- They generate data by AI, and evaluate results also by AI, which makes a huge possbility of bias exsiting. And there are few discussions about how to monitor and judge it.
- They seem just use one type of AI, which makes the knowledge distributed not evenly.

W4. Authors use many public resources as data resources, but I do not see any license introduction about it, did I miss anything?

[1]. Are Large Language Models Ready for Multi-Turn Tabular Data Analysis? ICML'25 \
[2]. InfiAgent-DABench: Evaluating Agents on Data Analysis Tasks ICML'24. \
[3] Text2Analysis: A benchmark of table question answering with advanced data analysis and unclear queries AAAI'24 \
[4] A Benchmark for Parsing Ambiguous Questions into Database Queries NeurIPS'24

**Questions:**

Q1: What is the **significant** difference between this work and [1]? Beyond the superficial difference in scope , what novel method contribution does this paper provide that isn't already in [1]?

Q2: The paper generates and evaluates data entirely through AI (GPT-4o for Curator/Reviewer agents, then GPT-4o for evaluation). This creates circular bias with minimal oversight. Unlike [1] which conducted comprehensive human expert validation, and [2-3] which discuss bias evaluation in AI-generated benchmarks, this paper lacks any serious bias analysis or quality assurance mechanisms. How can the authors claim benchmark reliability when the entire pipeline is AI-based with insufficient quality control?

Q3: The abstract states existing benchmarks don't capture "such complexities" referring to "vague and underspecified" goals, yet [1], [4] and others already address this. The abstract mentions three contributions but doesn't clearly connect them to solving the stated problem. Why should readers believe this identifies a gap that [1] hasn't already addressed more rigorously?

Q4:The paper uses "public datasets" as data sources for generating benchmarks but provides no license information or discussion of licensing compliance. Can you clarify the licensing status of all data sources and whether proper attribution/permissions were obtained?

**Details Of Ethics Concerns:**

Authors state that they use public data as resources as shown in Table 2, but I didn't find which license of these data. For example, in Kaggle, not all the data do they allow for distribution or built upon them.

---

### Official Review · Reviewer_cBLZ · 2025-10-30

**Soundness:** 3
**Presentation:** 3
**Contribution:** 2
**Rating:** 4
**Confidence:** 4

**Summary:**

This paper introduces ConDABench, a new framework for evaluating large language models (LLMs) on conversational data analysis (ConDA) tasks.
The authors argue that existing benchmarks fail to capture the complexity of real-world data analysis, which often involves ambiguous, under-specified user goals and unclean data that require interactive clarification.
ConDABench addresses this gap with two main components:
A Multi-Agent Generation Pipeline: This workflow creates 1,420 realistic ConDA problems by "reverse engineering" them from articles about public datasets. It generates a query-answer pair and, crucially, the specific Python code required to produce that answer from the data.
An Interactive Evaluation Harness: This tool evaluates models by using a novel "User Proxy Agent". When a model-under-test asks a clarifying question (e.g., "What threshold should I use for 'low-budget'?"), the User Proxy consults the ground-truth code to provide a consistent answer, allowing the conversation to proceed realistically without human intervention.




Using this benchmark, the paper finds that while newer LLMs are better at solving more problems, they are not necessarily better at tasks requiring sustained, long-form engagement. In fact, newer models often achieve higher accuracy with fewer conversational turns, suggesting they are being optimized to be better one-shot problem solvers rather than true collaborators.

**Strengths:**

1. The paper's core premise is excellent. It correctly identifies that real-world data analysis is messy, iterative, and interactive. Existing benchmarks that rely on static, one-shot question-answering fail to capture this complexity, especially regarding under-specified queries and unclean data. This work makes a serious attempt to solve that.

2. The multi-agent pipeline (Curator, Reviewer, Code Generator) for creating the benchmark from real-world articles is a very robust design. Using the generated code as a "proofing" step to ensure the query-answer pair is valid and grounded in the data is a key feature for ensuring benchmark quality.

**Weaknesses:**

1. Lack of Novelty in Motivation and Methodology

The primary weakness of this work is its significant overlap with existing literature [1]. The core motivation—that data analysis is interactive and involves handling ambiguous or “under-specified” queries—is not new and has already been a central focus in [1].
Workflow: The use of a multi-agent workflow to generate benchmark data is also found in [1].
Code-Grounding: The paper’s “code-grounded” method is presented as a key contribution. However, [1] also features a code generation track and produces question-answer pairs based on code execution. The running example in Figure 2, which focuses on clarifying “low-budget” and handling missing data, is conceptually very similar to the examples and motivations presented in [1].

2. Questionable Benchmark Quality and Rigor

The claims about the benchmark’s quality are not sufficiently substantiated, especially when compared to the rigorous validation practices in related work.

Insufficient Human Validation: The authors mention that the benchmark was validated by human experts, but this process is described in only one paragraph. The paper provides a single agreement number (92.73%) without explaining the rater instructions, evaluation criteria, or inter-annotator agreement metrics. In contrast, [1] includes detailed sections on its human evaluation protocol, making it difficult to assess the validity of the ConDABench dataset.

Absence of Bias Evaluation: The benchmark is generated using a multi-agent, LLM-based workflow, which is known to introduce or amplify biases. However, the paper does not discuss or evaluate this risk.

3. Unclear Contribution of the “User Proxy Agent”

The authors’ main claim to novelty appears to be the “User Proxy Agent.” However, the concept of a simulated user agent for evaluation already exists in [1]. The paper argues that its proxy is unique because it is grounded by reverse-engineered code to answer model clarifications (e.g., defining “low-budget”), but it is unclear how this is fundamentally different from the agent-based simulation in [1], which also uses generated code in its process. The paper would be significantly stronger if it clearly articulated the precise, novel aspects of its User Proxy beyond what has been established previously.


[1]. Are Large Language Models Ready for Multi-Turn Tabular Data Analysis? ICML25

**Questions:**

Q1 Potential for "Ground-Truth" Overfitting: The paper itself points out a serious methodological flaw in Figure 21. The Code Generator's objective is to produce code that matches the article's (potentially incorrect) answer. In the example, the code is shown to explicitly filter out the correct answer ("Solomon Islands") just to match the article's incorrect answer ("Afghanistan"). This means the "ground truth" code is not "correct reasoning" but "reasoning that replicates the article's (possibly flawed) conclusion." This is a significant concern for benchmark validity.

Q2 Reliance on GPT-4o as a Judge: The entire evaluation harness—both the User Proxy and the correctness/conversation quality Evaluators—is powered by GPT-4o. While the authors correctly avoid using GPT-4o as a contestant, this still anchors the entire benchmark's results to the performance, biases, and quirks of a single proprietary model. The paper's main findings (Figs. 5-7) are, in effect, "what GPT-4o thinks of other models' performance."

Q3 "Deep" vs. "Shallow" Definition is a Proxy: The paper defines task depth based on the number of iterations it took the generator-reviewer agents to create the code. This is a proxy for the difficulty of benchmark creation, not necessarily the intrinsic cognitive complexity of the task for the model-under-test. It's an interesting heuristic but may not be a reliable measure of reasoning depth.

---

### Official Review · Reviewer_cYCq · 2025-10-30

**Soundness:** 2
**Presentation:** 2
**Contribution:** 1
**Rating:** 2
**Confidence:** 4

**Summary:**

This paper introduces ConDABench, a framework for generating conversational data-analysis benchmarks and evaluating external tools on those benchmarks. The framework comprises: (1) a multi-agent pipeline that derives realistic tasks from articles describing insights on public datasets by automatically extracting question–answer pairs and back-synthesizing executable code to support the answers; (2) a dataset of 1,420 ConDA problems spanning open-ended analysis, projection/forecasting, and standard QA; (3) an interactive evaluation centered on a User Proxy Agent that converses with systems under test to measure both task correctness and dialogue quality. Experiments on state-of-the-art LLMs show that, although newer models solve more isolated instances overall, they are not necessarily better at tasks that require sustained, multi-turn collaboration.

**Strengths:**

1. The code‑grounded setup ties c to (q, a, d) so the User Proxy can answer clarifications consistently, making interactions closer to real workflows and providing a built‑in proof step via reverse‑engineering.

2. Data sources span TidyTuesday, Kaggle, and ScienceDirect with a purportedly uniform pipeline. The same tasks run across three execution frameworks, enabling apples‑to‑apples comparisons.

3. Beyond accuracy, ConvQ uses SAT/DSAT rubrics with a regressor aligned to human judgments, which is helpful for diagnosing conversation behaviors.

**Weaknesses:**

1. Lack of novelty, motivation and contribution are not clearly distinguished from prior work. The paper does not clearly separate its contributions from closely related benchmarks. The gap relative to listed baselines is not demonstrated with rigorous, side-by-side evidence, and the running example feels standard—so the “first systematic” claim reads stronger than what is shown.

2. The evaluation anchors on GPT-4o for both proxy and grader; Appendix E shows Qwen3-32B as a drop-in, but not as a full Proxy×Judge grid on the entire set, leaving room for model-family bias concerns.

3. The task structure is unbalanced, and the scope might be shallow. Table 2 reports only 103/1420 (~7%) deep tasks and 11 projection items; with an average 24.02 LOC and 2.01 s runtime, the suite may under-stress heavy wrangling, multi-table joins, or long-horizon analyses.

4. The stop-on-final-answer policy with no corrective turn can inflate ConvQ/efficiency for systems that answer early, and penalize models that prefer to clarify before committing.

5. Figs. 5–7 plot point estimates with reference lines but no confidence intervals, rerun variance, or significance tests, making the strength of “gaps” hard to gauge.

6. Many tasks admit multiple valid solutions; using one reverse-engineered code path as the sole reference risks penalizing equivalent alternatives.

7. Sec. 3.3 lists sources but not per-source licensing/redistribution details; public links appear in captions, which may be problematic during the blind-review period.

**Questions:**

1. The paper lists three contributions, but it does not clearly explain how they address the problem stated in the abstract. The abstract claims existing benchmarks 'do not capture these complexities'. Could the authors elaborate on how the paper's contributions specifically solve this problem?

2. Does the current stop-on-final-answer with no correction policy systematically favor early-answering systems? Under an alternative like one light correction or final-and-consistent stopping, how would ConvQ, turns, and accuracy shift?

3. Given there are only 103/1420 (~7%) deep tasks and 11 projection items, would the trends likely change if deeper cases or heavier wrangling/join scenarios were expanded?

4. For tasks with inherently multiple valid answers, do you define equivalence classes or tolerance bands? How is this implemented in the evaluation pipeline?

---

### Official Review · Reviewer_Gzgt · 2025-10-30

**Soundness:** 2
**Presentation:** 2
**Contribution:** 1
**Rating:** 2
**Confidence:** 4

**Summary:**

ConDABench presents a code-grounded benchmark for evaluating LLMs in conversational data analysis, where models can ask clarification questions before producing answers. It spans 1420 tasks built via a multi-agent pipeline and an automated evaluation harness. The claimed interactivity and novelty are limited, as prior benchmarks like InfiAgent-DABench and TAPilot-Crossing already explore similar multi-turn interactive analysis. Only 59% of curated QA pairs are valid, and benchmark reliability remains uncertain due to errors in the generated code. More critically, the evaluation pipeline depends on GPT-4o for both simulating users and grading, creating circularity and hindering reproducibility. Overall, despite strong engineering and scale, the work’s conceptual contribution and data integrity fall short of ICLR standards until it provides cleaner validation and an open, model-agnostic assessment setup.

**Strengths:**

ConDABench offers engineering contribution through its code-grounded, multi-agent generation pipeline, which automatically links natural-language queries to executable analysis code, ensuring transparent reasoning steps. It also provides a benchmark of 1420 data-analysis tasks spanning domains.

**Weaknesses:**

1. **Limited validation of benchmark faithfulness and correctness:** Although the authors validate 20% of samples and report a 92.73% correctness rate comparable to ImageNet, this verification is insufficient for a dataset explicitly designed for interactive reasoning. Errors in benchmark code (see Appendix O) can cascade through the evaluation harness, biasing results. Moreover, the validation metric only checks surface-level correctness (answer-code consistency), not semantic alignment or reproducibility across different code interpreters. Without reproducibility tests, it’s unclear whether results hold across Python environments or library versions.

2. **Over-reliance on GPT-4o for core evaluation components:** The entire evaluation pipeline depends on GPT-4o, including the User Proxy and Evaluator agent. This introduces model bias and circularity despite claims of “model-agnostic evaluation”. GPT-4o’s generation patterns (verbosity, preference for cautious disambiguation etc.) can inadvertently advantage or penalize certain models. Even though Appendix E compares GPT-4o to Qwen-3-32B and finds “close alignment”, a 6 - 8% difference is non-trivial. Without multi-model agreement metrics or human-in-the-loop calibration, it is difficult to ensure fairness and stability across model families.

3. **Incomplete coverage of real-world analytic behaviors:** The paper admits that ConDABench “does not cover exploratory tasks or user intent changes”. These are precisely the cases that define human–AI collaboration in data science. The current benchmark focuses on query–answer fidelity and short clarifications but omits behaviors like goal reformulation, intermediate hypothesis testing, or iterative visualization refinement. As a result, the benchmark may assess technical precision but not collaborative problem solving, which limits its external validity for “real-world DA assistants.”

4. **Lack of ablation or sensitivity analysis for pipeline design choices:** The multi-agent pipeline involves several components (Curator, Code Generator, Reviewer, Audited Reviewer), yet no ablation demonstrates their relative contributions. For instance, how many invalid (q, a) pairs were filtered by the Reviewer vs. the Code Generator? Or how does reviewer leakage quantitatively affect benchmark validity? Without such analysis, the claimed “robustness and extensibility” (in Sec. 3.3) remains questionable. This is critical since even minor prompt variations could change outcomes.

**Questions:**

Since the user proxy references code to answer clarifications, how is conversational naturalness preserved?

---

### Note · Authors · 2025-11-17

I have read and agree with the venue's withdrawal policy on behalf of myself and my co-authors.